# Long-persisting SARS-CoV-2 spike-specific CD4$^+$ T cells associated with mild disease and increased cytotoxicity post COVID-19

Guihai Liu [1,2,10], Elie Antoun [1,2,10], Anastasia Fries [3], Xuan Yao[1,2], Zixi Yin[1,2], Danning Dong[1], Wenbo Wang[1], Peter A. C. Wing [1,2], Wanwisa Dejnirattisa[1,3], Piyada Supasa[1,3], Chang Liu [1,3], Timothy Rostron[4], Craig Waugh[5], Kevin Clark[5], Paul Sopp[5], Jeremy W. Fry [6], Iolanda Vendrell[1,7], Jane A. McKeating [1,7], Juthathip Mongkolsapaya[1,3,8], Gavin R. Screaton [1,3], Benedikt M. Kessler [1,7], Roman Fisher [1,7], Graham Ogg [1,9], Alexander J. Mentzer [1,3,11], Julian C. Knight [1,3,11], Yanchun Peng [1,2,9,11] & Tao Dong [1,2,9,11] ✉

The recent COVID-19 pandemic left behind the lingering question as whether new variants of concern might cause further waves of infection. Thus, it is important to investigate the long-term protection gained via vaccination or exposure to the SARS-CoV-2 virus. Here we compare the evolution of memory T-cell responses following primary infection with subsequent antigen exposures. Single-cell TCR analysis of three dominant SARS-CoV-2 spike-specific CD4$^+$ T-cell responses identifies the dominant public TCRα clonotypes pairing with diverse TCRβ clonotypes that associated with mild disease at primary infection. These clonotypes are found at higher frequencies in pre-pandemic repertoires compared to other epitope-specific clonotypes. Longitudinal transcriptomics and TCR analysis, combined with functional evaluation, reveals that the clonotypes persisting 3–4 years post initial infection exhibit distinct functionality compared to those that were lost. Furthermore, spike-specific CD4$^+$ T cells at this time point show decreased Th1 signatures and enhanced *GZMA*-driven cytotoxic transcriptomic profiles that were independent of TCR clonotype and associated with viral suppression. In summary, we identify common public TCRs used by immunodominant spike-specific memory CD4$^+$ T-cells, associated with mild disease outcome, which likely play important protective roles to subsequent viral infection events.

Coronavirus disease 2019 (COVID-19) was a worldwide pandemic caused by severe acute respiratory syndrome coronavirus 2 (SARS-CoV-2), resulting in global morbidity and mortality. Studies in acute and convalescent individuals demonstrated that T cell responses to the virus associate with reduced disease severity[1,2], suggesting the importance of T cell responses in the control and resolution of SARS-CoV-2 infection[3–10].

Multiple immunodominant SARS-CoV-2 spike protein-specific T cell epitopes have been identified and are frequently detected in individuals who have recovered from infection and following vaccination[11–16]. More broadly, T helper 1 (Th1) and T follicular helper (Tfh) cell subsets have been well studied in SARS-CoV-2 natural infection and vaccination. Spike-specific memory Tfh cells are reported to persist following natural infection[17] and vaccination[15,16], and

importantly, they associate with sustained anti-Spike antibody responses[12,18,19]. Spike-specific Th1 cells were also observed in both infection and vaccination[20–22], with increased Th1 responses associating with severe COVID disease[18], while vaccine-induced Th1 cell responses correlated with the frequency of CD8+ T cells[12]. In addition to the canonical helper CD4+ T cell subsets, cytotoxic CD4+ T cells (CD4+ CTL) with direct effector and antiviral activity have been reported in several human viral infections[23–28]. Cytotoxic SARS-CoV-2-reactive CD4+ T cells have been identified in hospitalised COVID-19 patients[29] and 2 years following initial infection[30], with a significant expansion in lung infiltrates in severe disease[31]. However, whether SARS-CoV-2-reactive CD4+ CTLs contribute to virus clearance or to immune-mediated pathology remains unclear.

Current technological advances combining population-specific single-cell transcriptomic profiling with T cell receptor (TCR) sequence analysis have enabled researchers to study the quality of T cell responses and their ability to control virus replication[6,32,33]. However, due to the low frequency of antigen-specific T cells in the periphery and the limited availability of experimental tools for human study, there are very few studies investigating the immunodominant T cell responses along with their corresponding TCR repertoire, memory T cell establishment and their evolution following subsequent antigen exposure. As the T cell receptor repertoire is highly diverse, our knowledge of immunodominant epitope responses and their corresponding T cell receptor repertoire, along with a functional phenotype, is missing in the literature. Nevertheless, this analysis is critical for our understanding of the formation, evolution and function of the T cell memory pool arising from primary infection and the impact of subsequent antigen exposures, whether that comes from re-infection or vaccination.

In this study, we focus on three conserved immunodominant spike-specific CD4+ T cell epitopes (spike$_{166–180}$, spike$_{751–765}$ and spike$_{866–880}$) previously identified in individuals who had recovered from SARS-CoV-2 infection and vaccination[13–15,34,35]. We examine their TCR usage, phenotypical and functional differences between 1–3 months of convalescent and 3–4 years following initial infection by ex vivo single-cell transcriptomic and TCR repertoire analysis, along with in vitro functional evaluation. We identify dominant public TCRα clonotypes associated with mild COVID-19 disease by integrating our data with publicly available single-cell RNA sequencing (scRNA-seq) datasets from early pandemic cases. Our longitudinal analysis reveals that the clonotypes persisting after the initial infection show distinct functionality compared to clonotypes that are lost over time. Furthermore, transcriptional profiling uncovers key differences in Th1 and cytotoxicity signatures between 1–3 months and 3–4 years after initial infection. Notably, the increased cytotoxicity of CD4+ spike-specific T cells at 3–4 years is TCR independent, driven primarily by *GZMA* and associated with significant viral suppression. These findings provide critical insights into the long-term memory response to a previously unknown pathogen.

## Results

### Identification of immunodominant CD4+ T cell responses to SARS-CoV-2 spike protein
We and others previously identified three dominant SARS-CoV-2 spike protein (S) CD4+ T cell epitopes: S$_{166–180}$ (CTFEYVSQPFLMDLE)[13,14,16], S$_{751–765}$ (NLLLQYGSFCTQLNR)[13,15] and S$_{866–880}$ (TDEMIAQYTSALLAG)[13,34,35]. The HLA-restriction of these epitopes was defined using IFN-γ ELISPOT or peptide-MHC-Class II tetramer staining (Supplementary Fig. 1a–c). S$_{166–180}$-specific T cells are restricted by HLA-DPB1*04:01, while S$_{751–765}$- and S$_{866–880}$-specific T cells are restricted by HLA-DRB1*15:01. An overview of the study design can be seen in Fig. 1a. Our cohort comprised 48 individuals who had recovered from COVID-19 (Supplementary Data 1), with 30 (66.7% of 45) being HLA-DPB1*04:01 positive; and 17 (36.2%

of 47) carrying HLA-DRB1*15:01 (Fig. 1b). Ex vivo IFN-γ ELISpot analysis using convalescent PBMC samples showed that 68% (17/25) of DPB1*04:01 individuals responded to S$_{166–180}$, while 85.7% (12/14) and 71.4% (10/14) of HLA-DRB1*15:01 positive patients showed responses to S$_{751–765}$ and S$_{866–880}$ respectively (Fig. 1c), confirming the immunodominance of these epitopes.

To further characterise the immune response to these immunodominant epitopes, we generated 50 S$_{166–180}$-specific T cell clones from four participants, 54 S$_{751–765}$-specific T cell clones from four participants and 49 S$_{866–880}$-specific T cell clones from three participants. All those clones were established from 1–3-month convalescent samples, and the TCR clonotype was evaluated. Purity of the T cell clones was confirmed with tetramer staining after each round of expansion, and functional assays were only performed when purity was >95% (Supplementary Fig. 1d–f). To assess antigen-sensitivity of the clones, T cells were co-cultured with B cell lines loaded with titrated peptide and cytokine expression was measured by intracellular cytokine staining (ICS). Antigen-sensitivity of 32 S$_{166–180}$-specific, 45 S$_{751–765}$-specific and 48 S$_{866–880}$-speciifc T cell clones was evaluated. All clones expressed TNF-α, IFN-γ and IL-2 following antigen stimulation (Supplementary Fig. 1g). S$_{866–880}$-specific T cells showed the highest antigen-sensitivity, with the lowest half maximal effective concentration (EC$_{50}$) calculated from TNF-α, IFN-γ and IL-2 production.

### Dominant TCRα but broad TCRβ clonotypes amongst spike-specific CD4+ T cells
We carried out ex vivo SmartSeq2 to analyse the gene expression and the TCR of spike-specific CD4+ T cells from individuals at both 1–3 months and 3–4 years after infection (Supplementary Table 1). In total, our dataset comprised 702 tetramer-sorted cells from six patients during 1–3 months of convalescence (68 S$_{166–180}$-specific, 293 S$_{751–765}$-specific and 341 S$_{866–880}$-specific CD4+ T cells) and 1735 tetramer-sorted cells from ten patients at the 3–4-years follow-up (1135 S$_{166–180}$-specific, 266 S$_{751–765}$-specific and 334 S$_{866–80}$-specific CD4+ T cells). A further 379 cytokine-sorted S$_{166–180}$-specific T cells from five patients at 1–3 months of convalescence were analysed for their TCR repertoire.

Consistent with previous reports[16,29,36,37], we observed a high diversity in V gene usage for both the alpha and beta chains, with a similar broad repertoire at both timepoints (Fig. 2a, b). Interestingly, despite a relatively broad usage of TRAV genes, we identified a bias of TRAV35 used by S$_{166–180}$-specific cells (66.1% of all S$_{166–180}$-specific cells), TRAV12-1 used by S$_{751–765}$-specific cells (45.5% of all S$_{751–765}$-specific cells) and TRAV26-1 used by S$_{866–880}$-specific cells (41.5% of all S$_{866–880}$-specific cells) (Fig. 2a). Unlike the alpha chain, TRBV gene usage appears more diverse across all three epitopes, showing no specific bias (Fig. 2b). Investigating the beta chain pairing with the dominant TRAV genes shows a broad beta gene pairing for S$_{166–180}$-specific cells with negligible differences between timepoints (Fig. 2c). However, dominant TRAV genes for S$_{751–765}$- and S$_{866–880}$-specific T cells show a different preference in beta gene pairing between the timepoints: for S$_{751–765}$-specific cells, TRBV24-1 is preferential at 1–3 months whereas TRBV6-1 is dominant at 3–4 years; for S$_{866–880}$-specific T cells, TRBV20-1 pairs dominantly with TRAV26-1 at convalescence, while TRBV3-1 is preferred at 3–4 years (Fig. 2c, d). This finding was similar across all the participants and was not impacted by a particular individual. Overall, we found no difference in TCR clonotype diversity between the two timepoints other than TCRα clonotype diversity of S$_{166–180}$-sepcific T cells decreased at 3–4 years (Supplementary Fig. 2a, b).

An earlier study reported that TCRs recognising the same antigen have similar TCR sequences[38]. We therefore analysed similarity networks for the spike-specific TCR CDR3 alpha clonotypes and identified clusters of highly similar clones (Fig. 2d). A cluster defined as TRAV35-

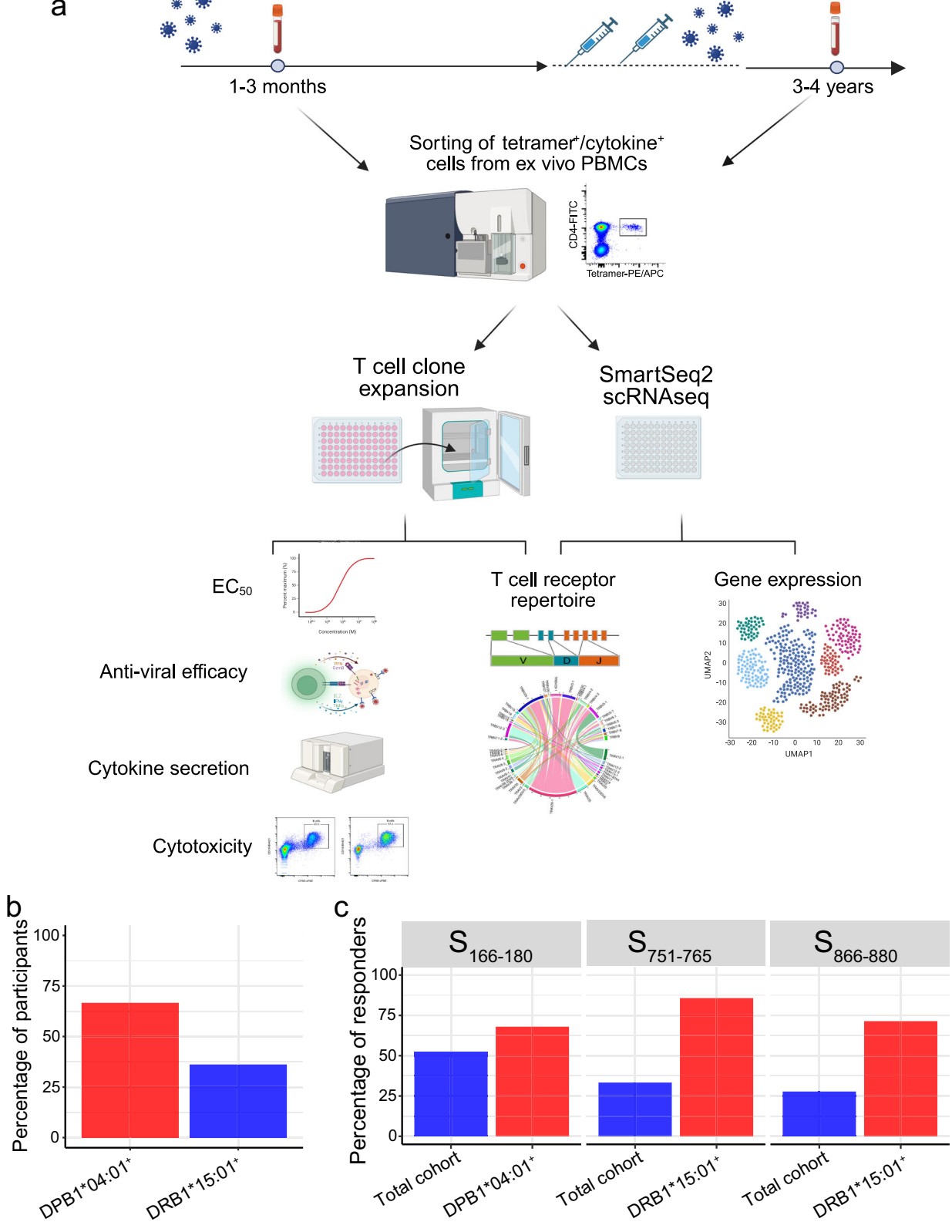

**Fig. 1 | Characterising three immunodominant spike-specific T cell responses targeting $S_{166-180}$-DPB1*04:01, $S_{751-765}$-DRB1*15:01 and $S_{866-880}$-DRB1*15:01 epitopes in COVID-19 patients. a** Overview of sample collection and study design. Created in BioRender. Dong, T. (2025) https://BioRender.com/r48m616.

**b** Proportion of patients with HLA-DPB1*04:01 ($n = 30/45$) and DRB1*15:01 ($n = 17/47$) in overall cohort. **c** Frequency of convalescent COVID-19 patients with T cells responding to $S_{166-180}$ ($n = 21/40$), $S_{751-765}$ ($n = 12/37$) or $S_{866-880}$ ($n = 10/36$) peptide stimulation.

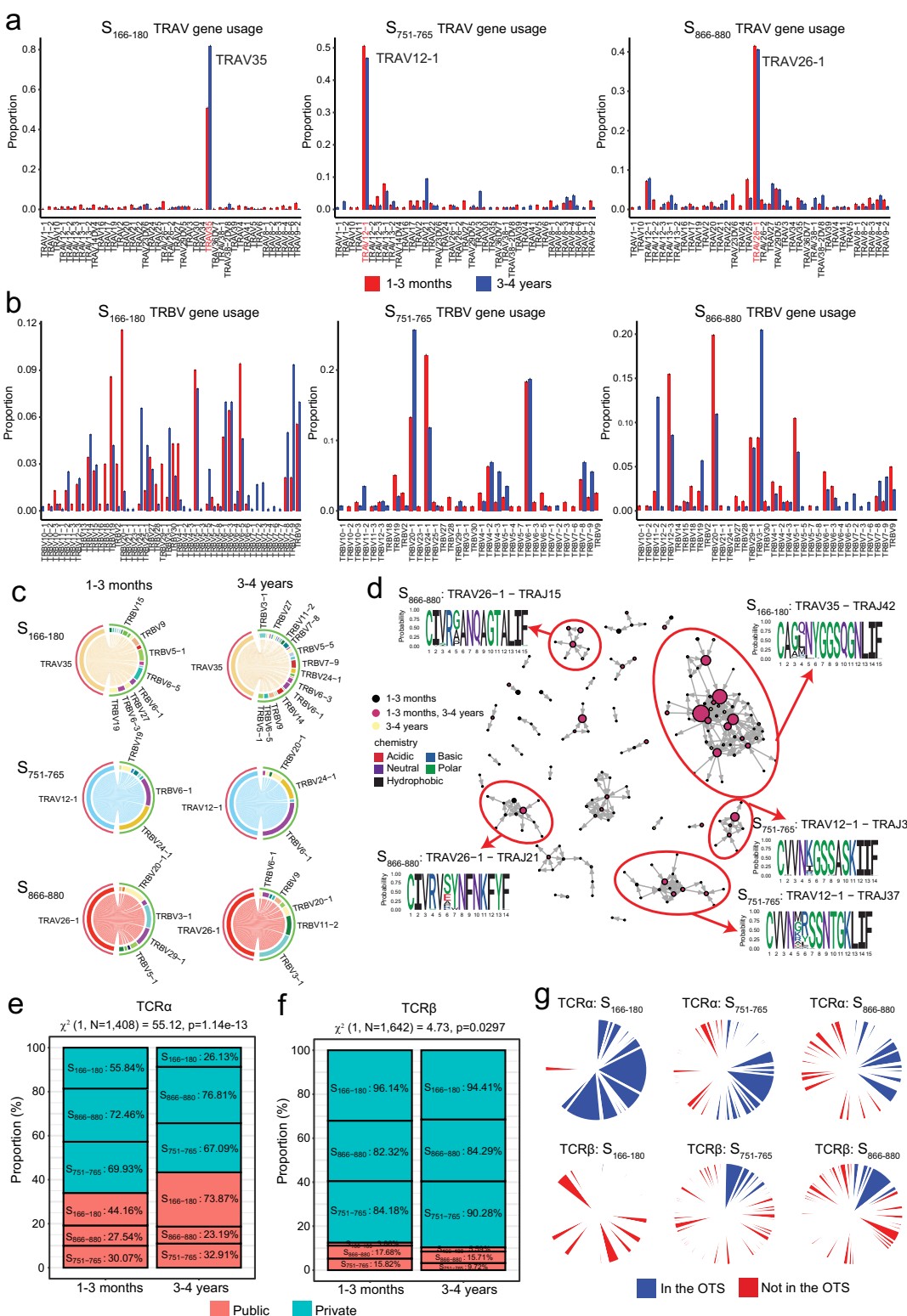

CAG**X**NYGGSQGNLIF, specific to $S_{166-180}$, is the largest alpha clonotype. This clonotype corresponded to the dominant TRAV gene used by the $S_{166-180}$-specific T cells, and is highly public, reported previously[16,36]. $S_{751-765}$-specific (TRAV12-1-CVVN**XX**SSNTGKLIF) and $S_{866-880}$-specific (TRAV26-1-CIVR**X**ANQAGTALIF) clusters were also identified, corresponding to the dominant TRAV genes used by $S_{751-765}$- and $S_{866-880}$-specific T cells, respectively. Similarity network

analysis of spike-specific TCR CDR3 beta clonotypes identified smaller clusters compared to the alpha clonotypes (Supplementary Fig. 2c), and these were used for downstream analysis.

We next sought to identify public TCR clonotypes, which are unique clonotypes shared among more than one unrelated individual. We identified 30 public alpha chains specific to $S_{166-180}$, 12 to $S_{751-765}$ and 14 to $S_{866-880}$, as well as 13 public beta chains specific to $S_{166-180}$, 8

**Fig. 2 | TCR repertoire analysis of spike-specific CD4$^+$ T cells at 1–3 months and 3–4 years post infection. a, b** TRAV (**a**) and TRBV (**b**) gene usage of $S_{166–180}$-, $S_{751–765}$- and $S_{866–880}$-specific T cells at 1–3 months (red bars) and 3–4 years (blue bars) post primary infection. **c** TRBV genes pairing with the dominant TRAV gene for $S_{166–180}$- (TRAV35), $S_{751–765}$- (TRAV12-1) and $S_{866–880}$- (TRAV26-1) specific cells at 1–3 months (left) and 3–4 years (right). **d** TCRα clonotype similarity network. Each vertex corresponds to an individual TCR clonotype, with edges connecting vertices if the CDR3 amino acid sequences show a normalised edit distance >0.9 (scRepertoire). The size of the vertex corresponds to the TCR clonotype frequency, and colour represents the timepoint at which they are found. Cluster motifs were generated using ggseqlogo and amino acid colours based on their biochemical properties. **e** The proportion of TCRα clonotypes classified as public (coral) or private (cyan) in our dataset. Percentages denote the proportion of cells of a particular epitope specificity classed as public or private. $\chi^2$ test of independence was used to compare proportions between 1–3 month and 3–4 years, with two-sided p-values calculated. **f** Proportion of TCRβ clonotypes classified as public (coral) or private (cyan) in our dataset. Percentages denote the proportion of cells of a particular epitope specificity classed as public or private. $\chi^2$ test of independence was used to compare proportions between 1–3 months and 3–4 years, with two-sided $p$ values calculated. **g** Pie charts denoting the proportion of TCRα (top) and TCRβ (bottom) clonotypes found in the Observed TCR space (OTS).

to $S_{751–765}$ and 8 to $S_{866–880}$ (Supplementary Data 2). Comparing the proportion of public clonotypes at different timepoints, we observed an increase in the proportion of cells with a public TCRα clonotype at 3–4 years compared to 1–3 months ($\chi^2 = 55.12$, $p = 1.14 \times 10^{-13}$ [Fig. 2e]), with 73.87% of $S_{166–180}$-specific T cells having a public TCRα clonotype at 3–4 years. We did not observe a significant correlation between the number of vaccine doses and the proportion of cells with a public TCRα clonotype at 3–4 years ($p = 0.9049$). Moreover, we found no evidence of preferential retention of public TCRα clonotypes in specific individuals; all participants exhibited cells with public TCRα clonotypes. In contrast, the TCRβ clonotype showed a much lower overall proportion of cells with a public TCRβ clonotype (Fig. 2f), although there is a slightly greater proportion of cells with a public TCRβ clonotype at 1–3 months compared to 3–4 years ($\chi^2 = 4.73$, $p = 0.03$ [Fig. 2f]). To confirm the public nature of these clonotypes and eliminate the potential risk of cross-contamination, we used the observed TCR space (OTS) database[39], which includes 3,185,982 paired TCRS from 892 individuals reported in 13 independent COVID-19 studies. 84.1% of the alpha clonotypes and 62.1% of the beta clonotypes classified as public in our dataset were also identified in the OTS. Using the OTS, we identified a total of 230 unique public alpha clonotypes and 75 unique public beta clonotypes (Fig. 2g).

## Dominant TCRα clonotypes are associated with mild COVID-19 disease during primary infection

To investigate whether these public dominant TCRα clonotypes are associated with COVID-19 disease severity, we utilised two publicly available datasets of SARS-CoV-2-reactive CD4$^+$ T cells with available TCR information, identified based on the upregulation of CD154/CD137/CD69 expression following peptide pool stimulation, for 52 COVID-19 patients[29,37] (Supplementary Table 2). Cells were classified as having a dominant TCRα clonotype if their CDR3α matches the identified clonotypes, TRAV35 ($S_{166–180}$), TRAV12-1($S_{751–765}$) and TRAV26-1($S_{866–880}$) shown in Fig. 2d, whereas the rest were considered as non-dominant. For both the ref. 29 and the ref. 37 datasets, we identified a significant association between having the dominant TCRα clonotype and mild COVID-19 disease ($\chi^2 = 226.17$, $p = 7.72 \times 10^{-50}$ [Fig. 3a] and $\chi^2 = 135.34$, $p = 4.08 \times 10^{-30}$ [Fig. 3b] respectively). Furthermore, since these two datasets used the same method to identify SARS-CoV-2-reactive CD4$^+$ T cells, combining them allowed us to find a significant association of the dominant TCRα clonotype to the mild disease outcome ($\chi^2 = 488.98$, $p = 6.59 \times 10^{-107}$, Fig. 3c). To confirm this association was not driven either HLA-DPB1*04:01 or HLA-DRB1*15:01-specific associations, we split the analysis to compare the proportion of cells with the dominant $S_{166–180}$-specific TCRα clonotypes compared to non-dominant in HLA-DPB1*04:01$^+$ individuals, and compared the proportion of cells with the dominant $S_{751–765}$-specific or $S_{866–880}$-specific TCRα clonotypes compared to non-dominant in HLA-DRB1*15:01$^+$ individuals (Supplementary Fig. 3). The dominant $S_{166–180}$-specific TCRα clonotypes in HLA-DPB1*04:01$^+$ individuals are associated with mild COVID-19 disease (Supplementary Fig. 3a), while the dominant $S_{751–765}$-specific or $S_{866–880}$-specific TCRα clonotypes in HLA-DRB1*15:01$^+$ participants are associated with non-hospitalised

COVID-19 (Supplementary Fig. 3b). Therefore, the difference in disease severity proportion is likely not driven by any particular epitopes. In summary, these data suggest that these dominant TCRα clonotypes play an important role in protecting individuals from developing severe disease.

Given the highly expanded and public nature of these dominant TCRα clonotypes, we explored the possibility that these TCRs may arise from sequences that are present at a higher-than-average frequency in a naïve, pre-pandemic repertoire. Spindler et al.[40] carried out paired TCR sequencing of six healthy individuals prior to 2020 and identified 513,963 unique TCRα clonotypes, with an average of 85,660 clonotypes per individual. From this dataset, we calculated the frequency of the dominant and non-dominant TCRα clonotypes. The dominant $S_{166–180}$-specific and $S_{866–880}$-specific TCRα clonotypes appeared at a significantly higher frequency in the pre-pandemic repertoires compared to the non-dominant clonotypes ($p = 0.042$ and $p = 0.033$ for $S_{166–180}$ and $S_{866–880}$ respectively, Fig. 3d). The dominant $S_{751–765}$-specific TCRα clonotype was detected at a higher frequency than non-dominant clonotypes in the pre-pandemic repertoires, although this did not reach statistical significance.

As SARS-CoV-2-specific responses have been reported to possibly originate from cross-reactive CMV-specific T cell[41] and pre-existing cross-reactive CD4$^+$ T cell responses[37], we cross checked our TCR clonotypes against those reported in the Immune Epitope Database (IEDB) and VDJ database (VDJdb). None of the public TCRs identified in this study were previously reported to be reactive against other viruses on an HLA-matched background.

## TCRα clonotypes maintained at 3-4 years show distinctive cytokine secretion

Given the longitudinal nature of our dataset, we examined whether clonotypes present during the convalescent period persisted for 3–4 years, or whether they are lost and replaced by new clonotypes, due to repeated vaccinations and infection with SARS-CoV-2 variants.

At 3–4 years, 80% of the $S_{166–180}$-specific cells retain a TCRα clonotype present at 1–3 months, predominantly the expanded and dominant public TCRα clonotype of TRAV35-CAG**X**NYGGSQGNLIF (Fig. 4a left); 46.2% of the $S_{751–765}$-specific cells have a TCRα clonotype found at 1–3 months, of which 65.8% are the dominant TCRα clonotypes of TRAV12-1-CVVN**XX**SSNTGKLIF and TRAV12-1-CVVN**X**GSSASKIIF (Fig. 4a middle); and 49.3% of the $S_{866–880}$-specific cells have a TCRα clonotype found at the 1–3-month timepoint, of which 50% are the dominant TCRα clonotypes of TRAV26-1-CIVR**X**ANQAGTALIF and TRAV26-1-CIVR**X**[Y/W]NFNKFYF (Fig. 4a right).

In contrast to the TCRα clonotypes, at 3–4 years, a lower frequency (10%) of $S_{166–180}$-specific cells have TCRβ clonotypes present at 1–3 months when compared to two other epitope-specific T cells, with 44.4% of the $S_{751–765}$-specific cells and 45.7% of $S_{866–880}$-specific T cells with a TCRβ clonotype also found at 1–3 months (Supplementary Fig. 4a). We next looked at the TCRβ clonotypes pairing with the previously identified dominant TCRα clonotypes (Fig. 4b). Similarly, TCRβ clonotypes pairing with the dominant $S_{166–180}$-specific TCRα clonotype

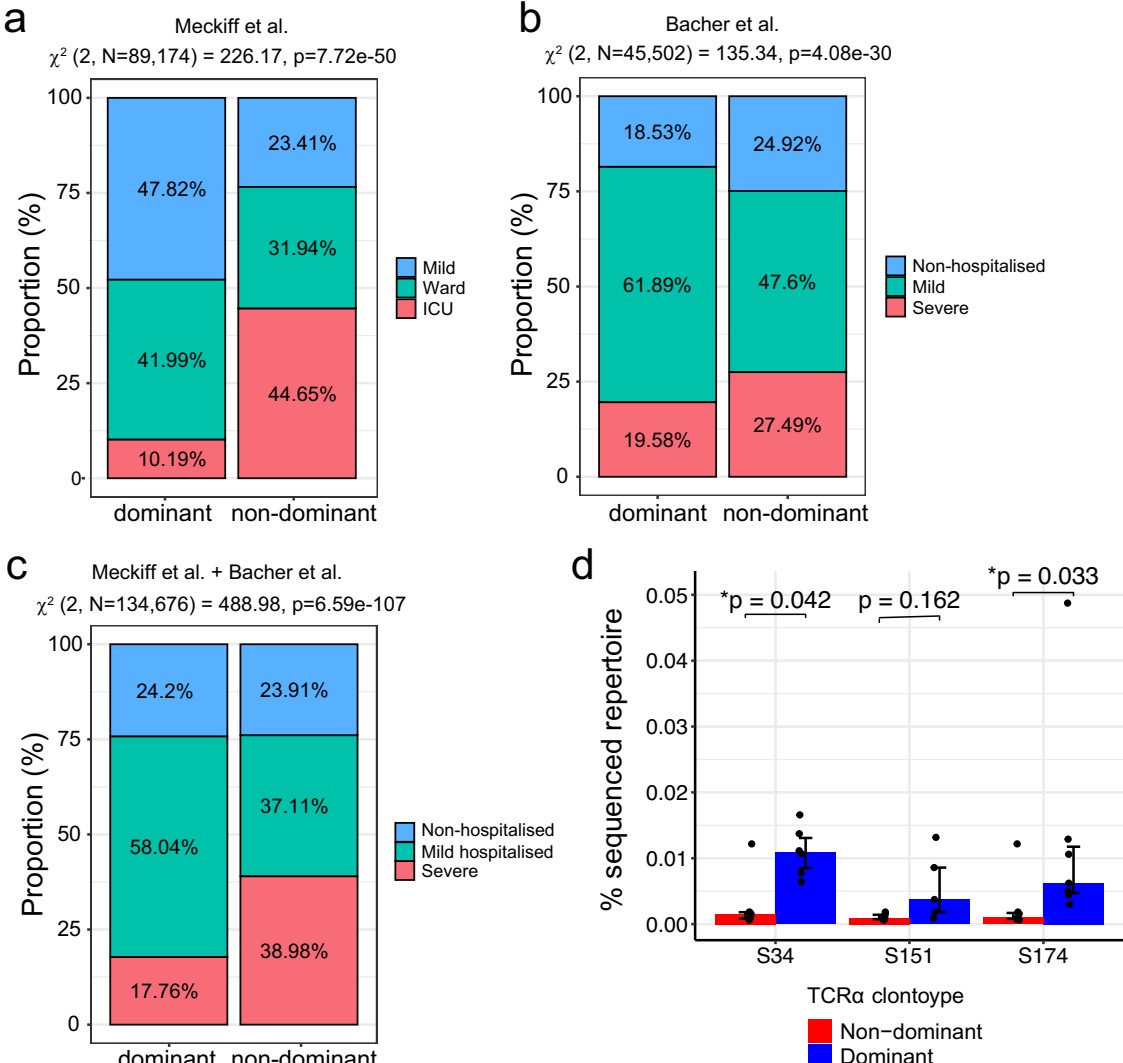

**Fig. 3 | Association between dominant TCRα clonotypes and disease severity. a, b** Frequency of cells from the Meckiff et al. dataset (**a**) and the Bacher et al. dataset (**b**) with dominant and non-dominant TCRα clonotypes split by COVID-19 disease severity of the donor. χ² test of independence was used to compare proportions between dominant and non-dominant TCRα clonotypes. χ² test of independence was used to compare proportions between dominant and non-dominant and two-sided *p* values calculated. **c** Frequency of cells from the Meckiff et al. and Bacher et al. datasets combined with dominant and non-dominant TCRα clonotypes split by COVID-19 disease severity of the donor. χ² test of independence was used to compare proportions between dominant and non-dominant TCRα clonotypes and two-sided *p* values calculated. Non-hospitalised COVID = Meckiff Mild + Bacher non-hospital; Mild hospitalised = Meckiff Ward + Bacher mild; Severe = Meckiff ICU + Bacher severe. **d** Percentage of the total TCR repertoire of dominant (blue bars) and non-dominant (red bars) TCRα clonotypes in pre-pandemic individuals (*n* = 6) from the Spindler et al. dataset. Plotted at median ± IQR. The Wilcoxon signed-rank test was used to compare between groups and two-tailed *p* values calculated.

show very little sharing between 1–3 months and 3–4 years, compared to S$_{751-765}$ and S$_{866-880}$-specific T cells, which show expanded TCRβ clonotypes paring with the dominant S$_{751-765}$- and S$_{866-880}$-specific TCRα clonotypes, maintained at 3–4 years.

As we identified clonotypes that were either maintained between the two timepoints sampled or not detected at the later timepoint, we interrogated our in vitro functional data of T cell clones isolated from 1–3-month samples to investigate whether the T cell clones with clonotypes maintained at the later timepoint showed differences in overall function. T cell clones were annotated as a maintained clonotype if their clonotype matched a TCR clonotype present in the single-cell TCR-seq dataset at both timepoints. Otherwise, T cell clones were categorised as either '1–3 month only' or '3–4 year only' if their TCR clonotype was found exclusively at one timepoint in the single-cell TCR-seq (Fig. 4c). Compared to the clones classified as '1–3 months

only', cytokine expression of S$_{751-765}$-specific T cell clones with maintained clonotypes showed significantly higher secretion of GM-CSF (adjusted *p* = 0.0014), IL-4 (adjusted *p* = 0.0014), IL-5 (adjusted *p* = 0.0042), IL-6 (adjusted *p* = 0.0042), IL-13 (adjusted *p* = 1.04 × 10⁻⁵) and TNF-α (adjusted *p* = 0.005) (Fig. 4d). However, when investigating the functional avidity of the S$_{751-765}$-specific T cell clones, no significant differences were observed in the IL-2 EC$_{50}$, TNF-α EC$_{50}$ or IFN-γ EC$_{50}$ between the two groups (Supplementary Fig. 4b, adjusted *p* > 0.05), suggesting that the clonal persistence may not be driven by functional avidity.

Taken together, these data suggest that spike-specific T cells with TCRα clonotypes detectable at both timepoints show differentiated cytokine profiles with higher expression of Th2 cytokines, compared to the T cells with TCRα clonotypes that were not detected at the 3–4-year timepoint.

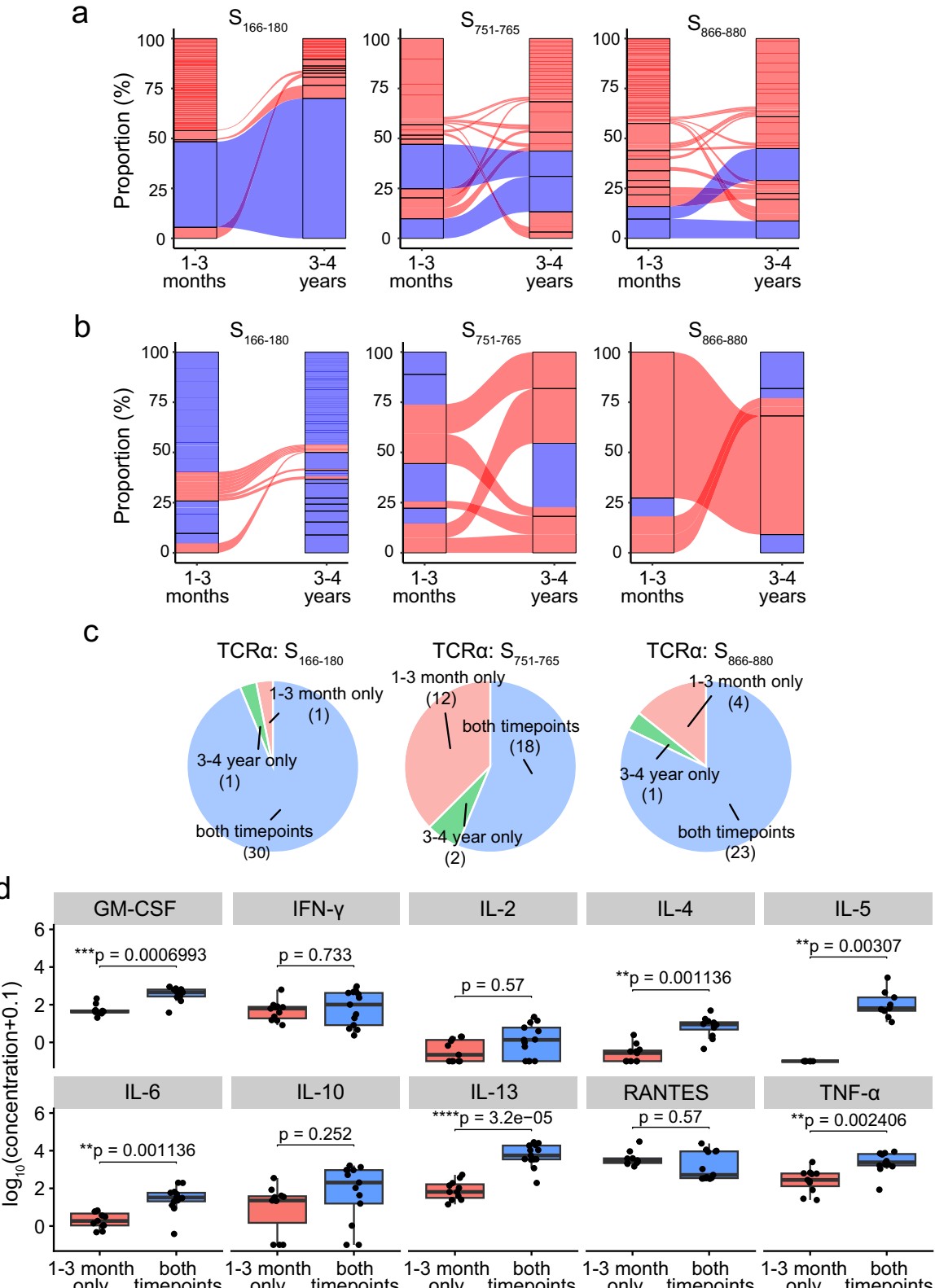

## Spike-specific CD4⁺ T cells show distinct transcriptional profiles over time

To investigate potential transcriptomic differences in the spike-specific T cells between the two timepoints, we analysed our scRNA-seq dataset that comprised 2213 total cells collected from two timepoints (1–3 months = 634 cells, 3–4 years = 1579 cells) across three epitope specificities (980 S$_{166-180}$-specific, 558 S$_{751-765}$-specific and 675

S$_{866-880}$-specific cells) from 13 individuals. Spike-specific T cells from the different individuals were integrated, and unsupervised clustering revealed ten distinct clusters based solely on their gene expression profiles (Fig. 5a, b). Interestingly, a unique cluster of cytotoxic T cells (CTL, cluster 7) exhibited high expression of cytotoxicity-related genes (e.g. *GNLY, GZMH, NKG7* and *GZMB*). In addition, cells in this cluster showed elevated expression of the apolipoprotein B mRNA editing

**Fig. 4 | Longitudinal TCRα and TCRβ analysis between 1–3 months and 3-4 years after infection. a** Alluvial plots highlighting the TCRα clonotypes at 1–3 months and 3–4 years, with links between columns denoting clonotypes found at both timepoints for $S_{166-180}$- (left), $S_{751-765}$- (middle) and $S_{866-880}$-specific (right) T cells. Blue links are the previously identified dominant TCRα clonotypes. **b** Alluvial plots highlighting the TCRβ clonotypes that pair with the previously identified dominant TCRα clonotypes at 1–3 months and 3–4 years, with links between the columns denoting clonotypes found at both timepoints. Red clonotypes are clonotypes found at both timepoints, whereas blue blocks are clonotypes found only at one timepoint. **c** Pie charts of the available clonal functional data based on whether the clone TCRα clonotype is found at both timepoints (blue), 1–3 month only (pink) or 3–4 year only (green) in the single cell data for $S_{166-180}$-specific (top), $S_{751-765}$-specific (middle) and $S_{866-880}$-specific (bottom) clones. **d** Luminex assay results showing the expression of ten cytokines released by $S_{751-765}$-specific clones split by whether the clones have a TCRα clonotype found at both timepoints ($n = 13$) or only found at 1–3 months ($n = 11$). Y-axis corresponds to $\log_{10}$[concentration (pg/ml) + 0.1]. Boxplots represent the 25th and 75th percentiles with the median marked with whiskers at ±1.5*IQR. The Wilcoxon signed-rank test was used for comparison between groups, and two-sided $p$ values were calculated.

enzyme, catalytic polypeptide (APOBEC) genes *APOBEC3C* and *APOBEC3G*. Other clusters that were identified included an interferon-stimulated genes (ISGs) positive cell cluster (cluster 10, marker genes include *OASL*, *IFIT2* and *ISG15*), two central memory clusters (clusters 5 and 1, with markers genes including *CCR7* [cluster 5] and *CXCR4* and *IL7R* [cluster 1]), and two clusters showing increased activation markers (clusters 4 and 6, with marker genes including *FOS*, *DUSP1* and *CD69* [cluster 4] and *ANXA1* and *LMNA* [cluster 6]). Clusters were generally comprised of cells from all patients and all sequencing runs (Supplementary Fig. 5a, b), indicating the clustering analysis does not represent patient-specific subpopulations or batch effects.

As scRNA-seq of $S_{166-180}$-specific cells at 1–3 months was conducted using cytokine-sorted cells and these cells were excluded from the longitudinal transcriptional analysis. We investigated whether there were any proportional differences in the number of cells from each timepoint across different clusters of $S_{751-765}$-specific and $S_{866-880}$-specific T cells. Although no differences were evident between these two epitope-specific cells, differences were observed when comparing 1–3-month and 3–4-year samples (Fig. 5c). At 3–4 years, the proportion of cells in clusters 1, 2 and 8 increased, while the cell populations at clusters 3, 4 and 6 declined compared to 1–3 months. This was confirmed by differential abundance testing using the miloR package (Fig. 5d, e).

To further investigate differences between memory cells from 1–3 months and 3–4 years, we generated module scores using gene lists compiled from the literature (Supplementary Table 3). Cells from 1–3-months exhibited a greater expression of genes involved in the positive regulation of Th1 differentiation (*ANXA1*, *CCR7* and *IRF1*), a greater level of integrin expression (*SELPLG*, *ITGAE* and *ITGA4*), and genes involved in T cell activation (*CD44*, *CD27* and *TUBA1B*) ($p < 0.0001$ for all, Fig. 5f). In contrast, cells from 3–4-year exhibited a greater cytokine gene expression (*IL16*, *IL32* and *CCL4*) and cytotoxicity (*GZMA* and *GNLY*) ($p < 0.0001$ for all, Fig. 5f). Moreover, this appears to be independent of epitope specificity (Supplementary Fig. 6a). Comparing the module scores for the $S_{751-765}$ and $S_{866-880}$ specific cells showed consistent differences, with increased Th1, Treg, integrin expression and activation signatures at 1–3 months and an increased cytotoxicity and cytokine signature at 3–4 years.

## Spike-specific CD4$^+$ T cells at 3-4 years exhibit increased cytotoxicity signatures driven by *GZMA* expression, correlating with viral suppression in vitro

Given the increased cytotoxicity signature observed at 3–4 years post infection, we further investigated whether this difference is driven by TCR clonotype. We categorised the cells into dominant or non-dominant TCRα groups. $S_{751-765}$- and $S_{866-880}$-specific CD4$^+$ T cells at 3–4 years consistently showed a greater cytotoxicity signature compared to cells at 1–3 months, regardless of whether the cells had a dominant or non-dominant TCRα clonotype (Fig. 6a). We confirmed this by examining clonotype sharing between cluster 7 (the CD4$^+$ CTL cluster) and the remaining clusters. Our analysis revealed that TCRα clonotypes are not restricted to cluster 7 but shared amongst all clusters (Fig. 6b), indicating that the difference in cytotoxicity is likely to be independent of TCR.

We then investigated individual genes driving the cytotoxicity signature at 3–4 years, and found that the expression of *GZMA* is significantly upregulated at 3–4 years (Fig. 6c, $p = 0.011$ and $p = 7.5 \times 10^{-8}$ for $S_{751-765}$- and $S_{866-880}$-specific cells, respectively), whereas other cytotoxic molecules, such as *GZMB* (Fig. 6d, $p > 0.05$) and *PRF1* (Fig. 6e, $p > 0.05$), showed no significant differences. This suggests that the cytotoxicity signature at 3–4 years is primarily driven by *GZMA*. Furthermore, at 3–4 years post infection, a greater proportion of cells in the cytotoxic cluster were found in $S_{166-180}$-specific T cells (CTL, $\chi^2 = 20.007$, $p = 4.52 \times 10^{-5}$ [Fig. 6f]), with 11.02% of cells being cytotoxic T cells, compared to 3.4% for $S_{751-765}$ and 5.68% for $S_{866-880}$. Amongst these cells, $S_{166-180}$-specific CD4$^+$ T cells express the highest level of *GZMA* at 3–4 years, compared to $S_{751-765}$- and $S_{866-880}$-specific cells (Fig. 6g, adjusted $p = 1.98 \times 10^{-6}$ and $2.64 \times 10^{-6}$ for $S_{166-180}$ vs $S_{751-765}$ and $S_{166-180}$ vs $S_{866-880}$, respectively). Given that our dataset contained three times as many $S_{166-180}$-specific cells as $S_{751-765}$/$S_{866-880}$-specific cells at 3–4 years, we randomly down-sampled the $S_{166-180}$-specific cells to 300 cells (matching the number of $S_{751-765}$- and $S_{866-880}$-specific cells), and still observed a higher level of *GZMA* expression in $S_{166-180}$-specific cells (Supplementary Fig. 6b).

This TCR-independence and *GZMA*-driven cytotoxicity was further validated in in vitro T cell clones specific for the three epitopes. We assessed the cytotoxicity of 98 CD4$^+$ T cell clones (33 $S_{166-180}$-specific CD4$^+$ T cell clones from four individuals, 30 $S_{751-765}$-specific clones from four individuals and 35 $S_{866-880}$-specific CD4$^+$ T cell clones from three individuals and all clones derived from 1–3-month samples). Clones were categorised as cytotoxic T cell clones if their killing capacity was >10% (see Methods for further details). Their killing of target cells was confirmed to be MHC class II-dependent by using an HLA-DR blocking antibody (Supplementary Fig. 6c). T cell clones with the same TCR clonotype exhibited distinct cytotoxic capability for all three epitopes, with some clones being cytotoxic and others not, despite the same TCR clonotype (Fig. 6h). Bulk RNAseq and proteomic analysis of three cytotoxic and three non-cytotoxic $S_{866-880}$-specific CD4$^+$ T cell clones showed the cytotoxic CD4$^+$ clones expressed higher *GZMA* at both RNA level and protein level, compared to non-cytotoxic clones (Fig. 6i, j, $p = 4 \times 10^{-4}$ and $p = 0.032$, respectively), with no significant differences in the *GZMB* expression. Importantly, cytotoxicity of the T cell clones associated with their suppression of SARS-CoV-2 replication (Fig. 6k, $\rho = 0.390$, $p = 7.05 \times 10^{-5}$), with cytotoxic T cell clones showing an overall greater level of viral suppression compared to non-cytotoxic CD4$^+$ T cell clones (Fig. 6l, $p = 0.00906$). These results suggest that spike-specific memory CD4$^+$ T cells at 3–4 years after infection retain more of a cytotoxic T cell memory, driven by *GZMA*, capable of controlling virus replication efficiently.

## Discussion

CD4$^+$ T cell memory plays an essential role in viral infections. However, the mechanisms underlying its long-term memory formation, particularly the phenotypical and functional changes, as well as the evolution of the TCR repertoire following repeated antigen exposures such as multiple vaccinations and re-infection, remain poorly understood in humans. This is largely due to the low frequency of antigen-specific T cells and the lack of suitable analytical tools. In this

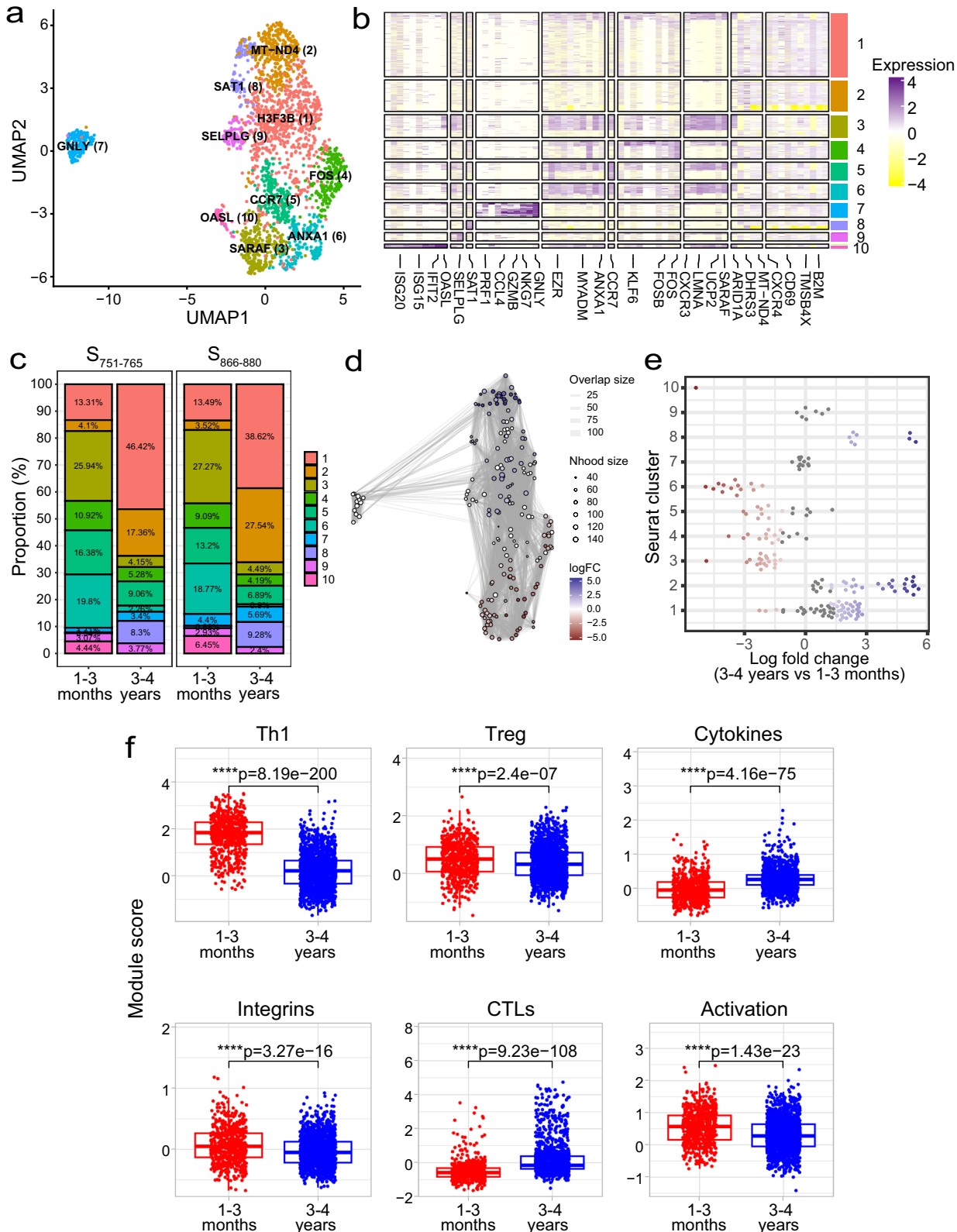

study, we longitudinally characterised the three most dominant spike-specific CD4$^+$ T cell responses restricted by common HLA Class II alleles (S$_{166-180}$-DPB1*04:01, S$_{751-765}$-DRB1*15:01 and S$_{866-880}$-DRB1*15:01), by interrogating the ex vivo TCR repertoires and transcriptomes of antigen-specific single cell, together with in vitro functional evaluation.

We firstly identified significantly expanded and dominant TCRα clonotypes in 1–3-month samples that are associated with mild COVID-

19 disease during the primary infection, when interrogating two studies from the early stages of the COVID-19 pandemic[29,37]. Given the limited number of participants in the two studies and the absence of HLA-typing for some individuals in Meckiff et al. study[29], these findings need to be further validated in larger cohorts. Nevertheless, public TCR clonotypes have been observed in the literature associated with dominant virus-specific responses[42]. Among the three epitopes, a higher frequency of S$_{166-180}$-specific T cells uses public TCRα

**Fig. 5 | scRNA-seq transcriptomic comparison of spike-specific CD4⁺ T cells at 1–3 months and 3–4 years. a** Uniform manifold approximation and projection (UMAP) visualisation of 2213 cells profiled ex vivo from PBMC samples. Cells are coloured based on their cluster occupancy. **b** Heatmap of the expression levels of the top differentially expressed genes for each cluster. **c** Stacked bar plots of the proportion of cells in each cluster split by timepoint. Shown here for $S_{751-765}$- and $S_{866-880}$-specific cells but not $S_{166-180}$-specific cells due to the lack of $S_{166-180}$-specific cells profiled at 1–3 months convalescence. **d** Milo analysis of differentially abundant cell clusters between 1–3 months and 3–4 years convalescence. Plot represents the embedding of the Milo differential abundance, where each node is a neighbourhood and node size is proportional to the number of cells in that neighbourhood. Colours represent the level of differential abundance. **e** Beeswarm plot showing the cell abundance changes between 3–4 years and 1–3 months convalescence. Neighbourhoods overlapping the same Seurat cluster identified in (**a**) are grouped together and neighbourhoods exhibiting significant differential abundance are coloured in red (higher at 1–3 months) or blue (higher at 3–4 years). **f** Boxplots comparing module scores for cells from 1–3 months ($n = 634$ cells) and 3–4 years ($n = 1579$ cells) convalescence. Boxplots represent the 25th and 75th percentiles with the median marked with whiskers at ±1.5*IQR. Wilcoxon signed-rank test was used to compare between groups and two-sided $p$ values calculated.

---

clonotypes, with more public TCRα clonotypes being identified. The expanded public TCR clonotype used by this epitope, together with the high prevalence of HLA-DPB1*04:01 at the population level[6], suggests that this response may contribute significantly to the CD4⁺ T cell memory response to SARS-CoV-2. Surprisingly, the dominant public TCRα clonotypes used by these three dominant spike-specific responses were found at a relatively high proportion in naïve pre-pandemic TCR repertoires compared to other epitope-specific clonotypes. Cross-reactive SARS-CoV-2-specific memory CD4⁺ T cells are readily detectable ex vivo in ~20–50 % of unexposed people[43,44]. The pre-existing cross-reactive T cells were induced by common cold viruses, or other coronaviruses, even other microbes, and have been reported to associate with mild COVID-19 infection[45–47]. Although it is not known whether the dominant and public TCRα clonotypes identified in our study are T cell precursors or pre-exiting cross-reactive TCRs, T cells bearing these TCRs may be able to elicit rapid T cell response during the primary infection, contributing to mild disease outcome.

We next examined the TCR repertoire changes of these three immunodominant spike-specific T cells between convalescence (1–3 months after primary infection) and 3–4-year follow-up. At 3–4 years, the majority of T cells retained the TCRα clonotypes found in 1–3 months, and the public dominant TCRα clonotypes of all three responses persist. T cell clones bearing the clonotypes detected at both timepoints show overall higher secretion of pro-inflammatory/Th2 cytokines such as GM-CSF, IL-4, IL-5, IL-6, IL-13 and TNF-α, when compared to the ones bearing clonotypes not detected over time, suggesting that CD4 T cells producing more Th2 cytokines during resolution stage are more likely to maintain at higher frequency in the long-term memory. Given that the sequences of $S_{166-180}$, $S_{751-765}$ and $S_{866-880}$ epitopes show very little similarity to those of seasonal coronaviruses but are highly conserved across SARS-CoV-2 variants, the seasonal coronaviruses are unlikely to have affected repeated recall of the spike-specific CD4 + T cells. However, repeated stimulations by SARS-CoV-2 reinfections or multiple vaccine doses potentially may have affected repeated recall of spike-specific T cells targeting the three epitopes, thus affecting their TCR repertoire, survival and cytokine profile at 3–4 years after primary infection.

Investigating the transcriptomes of these spike-specific cells, we unexpectedly identified a decreased Th1 signature, together with an increased cytotoxicity signature at 3–4 years compared to 1–3 months convalescence. Further analysis of the ex vivo single-cell data and in vitro functional and multi-omic data of T cell clones revealed that this transcriptome cytotoxicity change is TCR independent and driven by increased expression of *GZMA*. It is unclear how cytotoxic CD4⁺ T cells develop. Grey-Gaillard et al.[30] suggest that infection-associated inflammation may have imprinted a cytotoxic signature on memory CD4⁺ T cells, as they show that spike-specific CD4⁺ T cells induced by infection remained enriched for transcripts related to cytotoxicity, whereas spike-specific CD4⁺ T cells induced by mRNA vaccination did not. In our cohort, the participants were recruited at the convalescent stage, and then further sampled 3–4 years post primary infection. Between these two sampling points, these participants likely underwent multiple vaccinations and possible infections by SARS-CoV-2 variants. This repeated antigen stimulation may have contributed to the increased cytotoxicity signature observed at 3–4 years.

Unlike cytotoxicity of CD8⁺ T cells, which is predominantly mediated by perforin and granzyme B[48,49], the cytotoxicity of CD4⁺ spike-specific T cells appears to be driven by the increased expression of *GZMA*, with significantly higher expression of GZMA at 3–4 years, compared to 1–3 months, whereas no significant difference was observed in expression of perforin and GZMB between the two timepoints. The greater level of GZMA was confirmed at both the RNA and protein level in cytotoxic and non-cytotoxic $S_{866-880}$-specific T cell clones. These cytotoxic CD4⁺ T cell clones showed enhanced SARS-CoV-2 viral suppression in vitro, highlighting their direct effector function against virus replication[29,50]. Although it has been shown that direct killing of infected cells by CD4⁺ CTLs accumulated in the lung may have contributed to immunopathogenesis in severe COVID-19[29,50,51], these CD4⁺ CTLs in the long-term memory pool may contribute to protective immunity upon re-encountering the virus, which has been previously reported in an influenza human challenge[23].

Our study does have some limitations. Firstly, our analysis was carried out in a relatively small number of individuals. Therefore, the results we identified need to be validated and repeated in a larger number of individuals. Secondly, the analysis of the ex vivo spike-specific T cells by scRNA-seq is limited by their relatively low frequency in the circulation. Using tetramer staining to sort spike-specific CD4⁺ T cells from ex vivo PBMC samples has allowed for the analysis of highly specific single-cells, but limits the number of cells that can be analysed. Together with limited sampling depth, this may have resulted in insufficient TCR capture between the timepoints; therefore, the absence of certain TCR clonotypes at the later timepoint doesn't necessarily imply their loss over time. Furthermore, our study focused only on two timepoints, 1–3 months and 3–4 years after primary infection. Studying with sampling at additional timepoints in-between, particularly after each antigen-stimulation (vaccinations/breakthrough infections) would provide a detailed understanding of how the long-term memory pool evolved and established.

Taken together, in this study, we have identified high-frequency common public TCRs (in particular alpha chains) targeting three highly conserved immunodominant spike epitopes, which are associated with mild disease outcome, and increased cytotoxicity 4 years post initial infection, consistent with an important role in protecting the population from subsequent virus infection. Our data also provide new insights into understanding the evolution of memory T cell responses arising from human primary infection, the impact of subsequent antigen exposures, and potential intrinsic T cell factors that may contribute to differences in immune memory formation.

## Method
### Study participants and clinical definitions
Patients were recruited from the John Radcliffe Hospital in Oxford, UK, between March 2020 and September 2021 by the identification of patients hospitalised during the SARS-CoV-2 pandemic and recruited

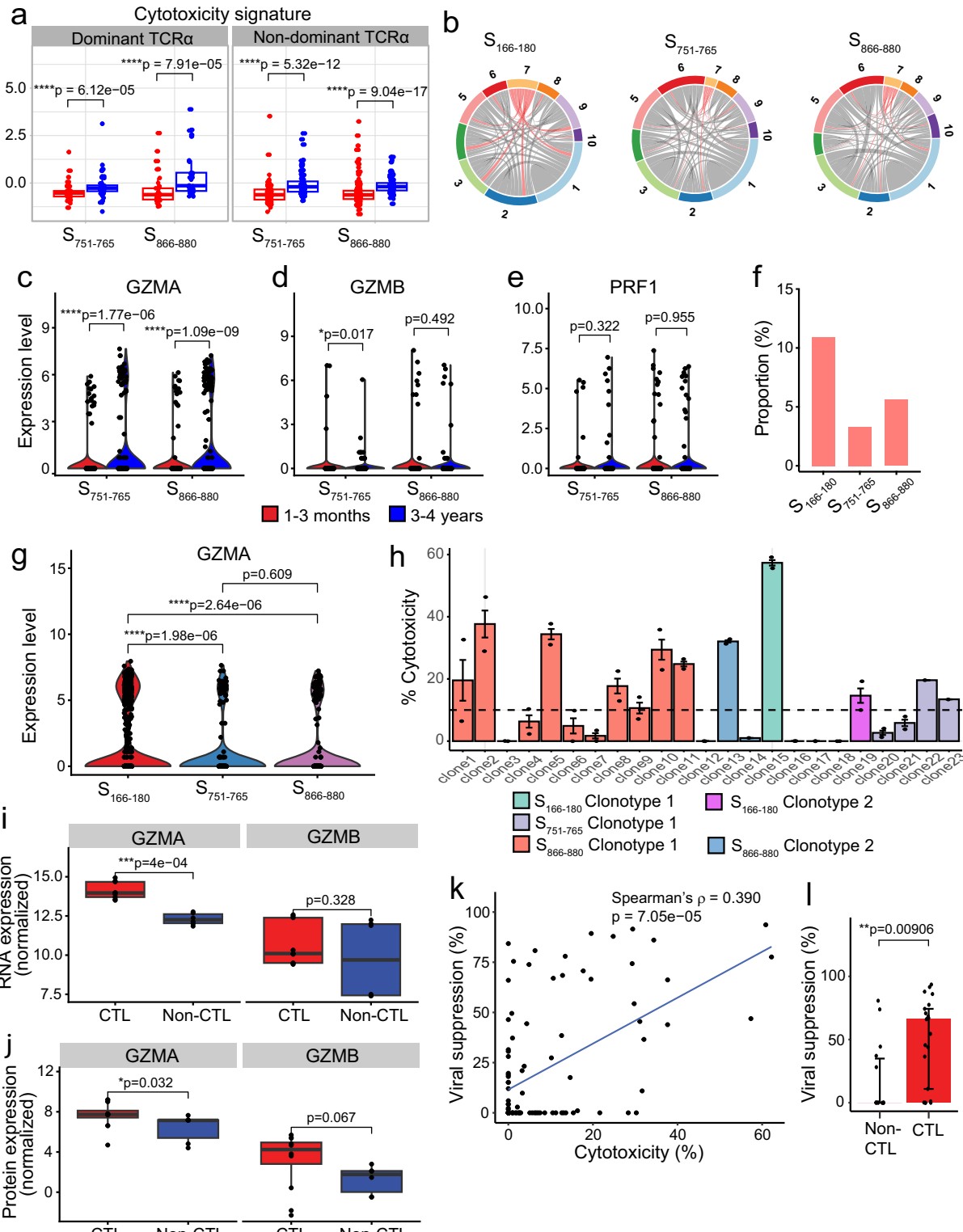

into the Sepsis Immunomics study. All participants were sampled at least 28 days after symptom onset during the primary infection. Ten of them, who were further sampled around 3–4 years after initial infection (Supplementary Table 1), had received multiple doses of vaccination and showed no symptoms of SARS-CoV-2 infection in 3–6 months before sampling (Supplementary Data 1). Written informed consent was obtained from all patients. Ethical approval was given by the South Central-Oxford C Research Ethics Committee in England (ref. 19/SC/0296). Clinical definitions were defined as previously described[52]. In brief, the degree of severity was identified as mild, severe or critical infection, according to recommendations from the World Health Organization.

**Fig. 6 | Investigation of the CD4[+] cytotoxicity signature in spike-specific cells at 3–4-year follow-up. a** Boxplots comparing the cytotoxicity signature of cells at 1–3 months (red) and 3–4 years (blue) convalescence for $S_{751-765}$- and $S_{866-880}$-specific cells, split by whether cells have a dominant TCRα clonotype or not ($n = 49$ [1–3 months] and 48 [3–4 years] cells for $S_{751-765}$-specific cells with dominant TCRα; $n = 33$ [1–3 months] and 34 [3–4 years] cells for $S_{866-880}$-specific cells with dominant TCRα; $n = 101$ [1–3 months] and 110 [3–4 years] cells for $S_{751-765}$-specific cells with non-dominant TCRα; $n = 166$ [1–3 months] and 104 [3–4 years] cells for $S_{866-880}$-specific cells with non-dominant TCRα). **b** Circos plots highlighting the clusters that share TCRα clonotypes found in cluster 7 (CD4[+] CTL cluster). **c–e** Violin plots comparing the expression of *GZMA* (**c**), *GZMB* (**d**) and *PRF1* (**e**) in cells between 1–3 months (red) and 3–4 years (blue). **f** Proportion of 3–4-year epitope-specific cells in the cytotoxic T cell cluster (CTL) compared to all other clusters (non-CTL). **g** Violin plots comparing the expression of *GZMA* in cells at 3–4 years between the three epitope-specific cells. **h** Cytotoxicity of $S_{166-180}$-, $S_{751-765}$- and $S_{866-880}$-specific T cell clones with the different TCR clonotypes. Each bar represents an individual T cell clone, and different coloured bars represent different clonotypes (3T cell clones with $S_{166-180}$ clonotype 1 and 2 clones with clonotype 2; 4 clones with $S_{751-765}$ clonotype 1; 12 clones with $S_{866-880}$ clonotype 1, and 2 clones with clonotype 2), plotted as median±IQR. **i, j** Boxplots comparing the expression of *GZMA* and *GZMB* in cytotoxic ($n = 3$ clones in triplicates) and non-cytotoxic ($n = 2$ clones in triplicates) CD4[+] T cell clones using bulk RNAseq (**i**) and bulk proteomics (**j**). **k** Correlation of virus suppression with T cell clone cytotoxicity ($n = 98$, including 33 $S_{166-180}$-, 30 $S_{751-765}$- and 35 $S_{866-880}$-specific CD4[+] T cell clones). **l** Comparison of virus suppression between $S_{866-880}$-specific CD4[+] cytotoxic ($n = 21$) and non-cytotoxic clones ($n = 14$), plotted as median ± IQR. The Wilcoxon signed-rank test was used to compare between groups (**a, c–g, l**), while a paired Wilcoxon signed-rank test was used for the bulk RNAseq and proteomic analysis in (**i, j**). Two-sided $p$ values were calculated for all Wilcoxon signed-rank tests. Correlation analysis was carried out using Spearman's correlation. All boxplots (**a, j, i**) represent the 25th and 75th percentiles with the median marked with whiskers at ±1.5*IQR.

## Generation of ACE2-transduced EBV-transformed B lymphoblastoid cell lines (BCLs)

EBV-transformed BCLs[53] and ACE2-transduced BCLs were established as described previously[6]. In brief, the cDNA for the human *ACE2* gene (ENSG00000130234) was cloned into a lentiviral vector backbone (Addgene plasmid ID 17488), then co-transfected with packaging plasmids pMD2.G and psPAX2 into HEK293-TLA using PEIpro (Polyplus) to produce lentivirus. EBV-transformed BCLs were infected with ACE2-coding lentivirus, followed by cell sorting via flow cytometry to enrich ACE2-expressing B cells. B cells with stable expression of ACE2 were maintained with 0.5 µg ml⁻¹ of puromycin (Thermo Fisher Scientific). *Mycoplasma* testing was carried out every 4 weeks with all cell lines using the MycoAlert detection kit (Lonza).

## Generation of T cell lines and clones

Short-term SARS-CoV-2-specific T cell lines were generated as described previously[53]. Briefly, $2 \times 10^6$ PBMCs were stimulated with 10 µM peptides at 37 °C for 1 h and cultured in H10 (RPMI 1640 medium with 10% human serum, 2 mM glutamine, 100 units/ml of penicillin and 100 µg/ml of streptomycin) at $2 \times 10^6$ cells per well in a 24-well plate (Costar). IL-2 was added to a final concentration of 100 IU/ml on day 3. $S_{751-765}$- and $S_{866-880}$-specific T cell clones were established by sorting tetramer[+] CD4[+] T cells from thawed PBMCs or short-term T cell lines on day 10-14. $S_{166-180}$-specific T cell clones were generated by cell sorting with TNF-α, IFN-γ, and IL-2 secretion assay (Miltenyi). T cell clones were then expanded with irradiated allogeneic PBMCs every 2–3 weeks as described previously[54].

## IFN-γ ELISpot assay

Ex vivo assays were carried out using either freshly isolated or cryopreserved PBMCs as described previously[13]. Peptides were added to $2 \times 10^5$ PBMCs at a final concentration of 2 µM for 16–18 h. For in vitro ELISpot assays, autologous and allogeneic EBV-transformed BCLs were loaded with peptides, and subsequently co-cultured with polyclonal T cell lines at an effector: target (E:T) ratio of 1:50 for at least 6 h, negative wells containing BCLs and T cells were included. To quantify antigen-specific responses, mean spots of the control wells were subtracted from the sample wells, and the results were expressed as spot-forming units (SFU) per 10⁶ PBMCs. Responses were considered positive if results were at least three times the mean of the negative control wells and >25 SFU/10⁶ PBMCs. If negative control wells had >30 SFU/10⁶ PBMCs or positive control wells (Phytohemagglutinin stimulation) were negative, the results were excluded from further analysis.

## Intracellular cytokine staining (ICS)

ICS was performed as described previously[13]. T cells were co-cultured with BCLs loaded with peptides at 37 °C for 6 h with GolgiPlug and GolgiStop (BD Biosciences). Cells were stained with Live/Dead Fixable Aqua dye (Invitrogen) followed by surface staining with CD4-PE-Cy7 (BD Biosciences). After subsequent permeabilisation with Fixation/Permeabilisation solution (BD Biosciences), cells were stained with IFN-γ-Alexa Fluor 488 (BD Biosciences), TNF-α-APC (eBioscence) and IL-2-BV421 (BioLegend). Negative controls without peptide were set up for each sample. Samples were run on Attune NxT Flow Cytometer (software v.3.2.1) and analysed using FlowJo v.10 software (FlowJo LLC). A representative gating strategy used for the ICS assay can be seen in Supplementary Fig. 7a.

## Cytokine production assessment

To assess cytokine production, T cells were co-cultured with BCLs loaded with or without peptide at an E:T ratio of 2:1. After 48 h, 50 µl of supernatant was collected for cytokine detection. Cytokines, including IFN-γ, TNF-α, IL-2, IL-4, IL-6, IL-10, IL-13, RANTES and GM-CSF were quantified using the Bio-Plex Pro Human Cytokine Assay (Bio-Rad) following the manufacturer's instructions, then ran on Bio-Plex 200 (Bio-Rad). Concentration was analysed by the Bio-Plex Manager (Bio-Rad).

## CFSE-based cytotoxic T lymphocyte killing assay

EBV-transformed BCLs were labelled with 0.5 µM carboxyfluoroscein succinimidyl ester (CFSE, Thermo Fisher Scientific), then loaded with 2 µM of peptide at 37 °C for 1 hour. Subsequently, cells were washed, counted and co-cultured with T cells at an E:T ratio of 4:1 at 37 °C for 6 h. Samples were then stained with 7-AAD (eBioscience) and CD19-BV421 (BioLegend). To assess the MHC class II-dependence of the killing, B cell lines were treated with either 40ug/ml of anti-HLA-DR antibody (BioLegend) or isotype control (BioLegend) at room temperature for 1 hour prior to being loaded with peptide. Cell death was assessed based on the presence of CFSE[+]CD19[+]7-AAD- (live) cells. Negative controls containing BCLs without peptide pulse and T cells were included for each sample. Samples were run on Attune NxT Flow Cytometer (software v.3.2.1) and analysed using FlowJo v.10 software (FlowJo LLC). A representative gating strategy used for the killing assay is shown in Supplementary Fig. 7b.

## Live virus suppression assay

As described previously[6], BCLs expressing ACE2 were infected with SARS-CoV-2 viruses (Victoria 01/20 strain, provided by J. McKeating[55]) at a MOI of 0.1 for 2 h at 37 °C. Cells were then washed and co-cultured with T cells at an E:T ratio of 4:1 at 37 °C. Control wells containing only virus-infected targets were included. Three replicates were set up for each condition. After 48 h incubation, cells were washed with PBS and lysed with RLT buffer (Qiagen), followed by RNA extraction using RNeasy 96 Kit (Qiagen). Virus copies were quantified by real-time qPCR using one-step RT mastermix kit (Eurogentec) and N1 probe contained in 2019-nCoV RUO kit (IDT), and the viral suppression rate was

calculated by the reduction of viral copies when antigen-specific T cells are present.

## Cell sorting for scRNA-seq

$S_{166-180}$-specific CD4$^+$ T cells from 1–3-month samples were sorted using a cytokine secretion assay following the manufacturer's instructions (Miltenyi Biotec). Briefly, $3–5 \times 10^6$ PBMCs were stimulated with $S_{166-180}$ peptide at a final concentration of 10 μM for 5 h. Subsequently, cells were washed and incubated with TNF-α, IFN-γ and IL-2 catching antibodies for 45 min, followed by staining with PE-conjugated TNF-α, IFN-γ and IL-2 detection antibodies, CD3-FITC, CD8-APC, CD14-PE-CF594, CD19-PE-CF594 and CD16-PE-CF594 (BD Biosciences), CD4-BV421 (BioLegend). Before sorting, cells were stained with PI (eBioscience) to exclude nonviable cells. $S_{751-765}$- and $S_{866-880}$-specific CD4$^+$ T cells from 1–3-month samples were sorted with peptide-MHC class II tetramers. In brief, $3–5 \times 10^6$ cells were stained with APC-conjugated HLA-DRB1*15:01 $S_{751-765}$ and $S_{866-880}$ tetramers (ProImmune), respectively. Live/dead fixable Aqua dye (Invitrogen) was used to exclude nonviable cells from the analysis. Cells were washed and stained with the following surface antibodies: CD3-FITC, CD4-PE (BD Biosciences), CD14-BV510, CD19-BV510, CD16-BV510 and CD8-BV421 (BioLegend). After exclusion of nonviable/CD14$^+$/CD19$^+$/CD16$^+$cells, CD3$^+$CD8$^-$CD4$^+$TNFα$^+$/IFNγ$^+$/IL-2$^+$ cells or CD3$^+$CD8$^-$CD4$^+$tetramer$^+$ were sorted for scRNA-seq using a BD FACSAria Fusion sorter or BD FACSAria III (BD Biosciences). Single cells were directly sorted into 96-well PCR plates (Thermo Fisher Scientific) containing cell lysis buffer and stored at −80 °C for further SmartSeq2 analysis. A representative gating strategy for the tetramer and cytokine sorting of single cells is shown in Supplementary Fig. 7c, d, respectively.

## Tetramer-associated magnet enrichment and cell sorting of spike epitope-specific CD4$^+$ T cells

$S_{166-180}$-, $S_{751-760}$- and $S_{866-880}$-specific CD4$^+$ T cells were enriched from 3- to 4-year samples prior to sorting for scRNA-seq, as previously described[56,57]. In brief, $1.5–3 \times 10^7$ PBMCs were labelled with APC- or PE-conjugated peptide-MHC class II tetramers ($S_{166-180}$ tetramer from NIH Tetramer Core Facility, $S_{751-760}$- and $S_{866-880}$ tetramers from ProImmune) for 30 min. Enrichment was then performed with anti-APC or anti-PE microbeads using magnetic-activated cell sorting technology (Miltenyi Biotec) following the manufacturer's instructions. Subsequently, enriched $S_{166-180}$-, $S_{751-760}$- and $S_{866-880}$-specific CD4$^+$ T cells were stained with CD3-BV786 and CD8-BV510 (BioLegend), CD4-FITC, CD14-PE-CF594, CD19-PE-CF594 and CD16-PE-CF594 (BD Biosciences). Before sorting, cells were stained with Propidium Iodide (PI) (eBioscience) to exclude nonviable cells. CD3$^+$CD8$^-$CD4$^+$tetramer$^+$ were sorted for scRNA-seq using a BD FACSAria Fusion sorter or BD FACSAria III (BD Biosciences).

## SmartSeq2 scRNA-seq

ScRNA-seq with ex vivo sorted TNF-α$^+$/IFN-γ$^+$/IL-2$^+$ or tetramer$^+$ cells was performed using SmartSeq2 analysis as described previously[5]. Reverse-transcription (RT) and PCR amplification were performed with the exception of using ISPCR primer with biotin tagged at the 5' end and increasing the number of cycles to 25. Sequencing libraries were prepared using the Nextera XT Library Preparation Kit (Illumina) and sequencing was performed on Illumina NextSeq sequencing platform with NextSeq Control Software v.4.

## Deep sequencing of the TCR repertoire of T cell clones

TCR usage of T cell clones was sequenced as described[6]. Total RNA was extracted from $5 \times 10^5$ cells of each clone using RNeasy Micro Kit (QIAGEN), and 100-300 ng of the RNA from each clone was used for the generation of full-length TCR repertoire libraries using SMARTer Human TCR a/b Profiling Kit/v2 (TAKARA) following the supplier's

instructions. After purification, libraries of all clones were pooled and sequenced using MiSeq reagent Kit v.3 (600 cycles) on a MiSeq (Illumina) with MiSeq Control Software v.2.6.2.1.

## Cell preparation for bulk RNAseq and proteomics

Three cytotoxic and three non-cytotoxic $S_{866-880}$-specific CD4$^+$ T cell clones were included for the analysis. T cells were thoroughly washed with PBS three times before RNA and protein extraction. Total RNA was extracted from $5 \times 10^5$ cells using the RNeasy Micro Kit (QIAGEN) for bulk RNAseq. To extract proteins for proteomics, $1 \times 10^6$ cells were lysed with 1% NP40 cell lysis buffer (Thermo Fisher Scientific), including 1X protease inhibitor cocktail (Sigma-Aldrich) and 1 mM phenylmethylsulfonyl fluoride (Thermo Fisher Scientific) on ice for 1 h, with vortexing every 10 min during the incubation. After cell lysis, the solution was centrifuged at $16,000 \times g$ for 10 min at 4 °C. Supernatant containing the proteins were transferred into new tubes and snap frozen on dry ice, followed by storage at −80 °C for further proteomic analysis. Three replicates from each clone were included for both RNAseq and proteomics.

## RNAseq library preparation

About 1 μg total RNA was sent to the Oxford Genomics Centre for total RNAseq analysis. Briefly, cDNA libraries were prepared using the NEB Ultra II Library Prep kit for Illumina (NEB) following the manufacturer's protocol and sequencing was performed on NovaSeq 6000 using a NovaSeq 6000 SP Reagent Kit v1.5 (300 cycles, Illumina).

## Proteomics sample preparation

Thirty micrograms of proteins were digested using S-trap micro spin columns following the manufacturer's protocol (Profiti). Briefly, SDS was added to the sample to a 2.5% final concentration. Samples were reduced with 10 mM DTT for 30 min and alkylated with 40 mM iodoacetamide for 30 min at room temperature in the dark. Samples were then acidified with phosphoric acid to a 1.2% final concentration, and proteins were precipitated by adding 90% methanol/100 mM TEAB buffer to a 1:7 ratio (sample:buffer). Samples were then transferred to the S-trap spin column and spun through with $4000 \times g$ and washed four times with 150 μl 90% methanol/100 mM TEAB ($4000 \times g$). About 20 μl of 50 mM TEAB containing 1 μg of trypsin (1:30 trypsin:protein ratio) was added to the S-trap spin column and incubated overnight at 37 °C. Peptides were then sequentially eluted with 50 mM TEAB, 2% formic acid and 2% formic acid in a 50% acetonitrile solution. Peptides were dried using a centrifugal evaporator.

## Liquid chromatography–mass spectrometry (LC-MS/MS)

Dried peptides were reconstituted in LC-MS/MS water containing 2% acetonitrile, 0.1% TFA. Thirty-three percent of tryptic peptides were analysed by liquid chromatography tandem mass spectrometry (LC-MS/MS) using Ultimate 3000 UHPLC (Thermo Fisher Scientific) connected to an Orbitrap Fusion Lumos Tribrid (Thermo Fisher Scientific). Briefly, peptides were loaded onto a trap column (Acclaim PepMax; 100 μm x 2 cm, nanoviper, Thermo Fisher) and separated on a 50cm-long EasySpray column (ES903, Thermo Fischer) with a gradient of 2–35% acetonitrile in 5% dimethyl sulfoxide, 0.1% formic acid at a 250 nl/min flow rate over 60 min. Eluted peptides were then analysed on an Orbitrap Fusion Lumos (instrument control software v3.3; Thermo Fisher). Data were acquired in data-independent mode as previously described by ref. [58]. Briefly, full scans (350–1650 m/z) were acquired in the Orbitrap with 120k resolution and maximum injection time of 20 ms, followed by 40 DIA scan windows covering full mass range from 361 to 1388 with variable widths adjusted to the precursor density as described by ref. [58]. MS2 scans were acquired in the Orbitrap between 200–2000m/z at a resolution of 30,000 and a normalised HCD collision energy set to 30%.

## Quantification and statistical analysis

### SmartSeq2 scRNA-seq data processing

BCL files were converted to FASTQ format using bcl2fastq v2.20.0.422 (Illumina). FASTQ files were aligned to the human genome hg19 using STAR v2.6.1 d. Reads were counted using featureCounts (subread v2.0.0). The resulting counts matrix was analyzed in R v4.0.1 using Seurat v4.0.1.

### Single-cell RNA sequencing analysis

Cells were filtered using the following criteria: minimum number of cells expressing specific gene = 3, minimum number of genes expressed by cell = 200 and maximum number of genes expressed by cell = 4000. Cells were excluded if they expressed more than 10% mitochondrial genes. Patient-specific cells were integrated using Harmony v.1.0 to remove batch effects. The AddModuleScore function (Seurat) was used to look at the expression of specific gene sets (Supplementary Table 3). Higher scores indicate that that specific signature is more highly expressed in a particular cell compared with the rest of the population. The FindMarkers function (Seurat) was used to evaluate differentially expressed genes (DEGs) between two conditions using the MAST (model-based analysis of single cell transcriptomics) statistical test, with different sequencing batches as latent variables. miloR[59] was used to carry out differential abundance testing between samples from 1–3 months and 3–4 years.

### TCR processing

TCR sequences were reconstructed from SmartSeq2 scRNA-seq FASTQ files using MiXCR v.3.0.13 to produce separate TRA and TRB output files for analysis. The output files were parsed into R using tcR v.2.3.2. Bulk TCR sequencing BCL files were converted to FASTQ files using bcl2fastq. TCRs were extracted using MiXCR and parsed into R as described earlier. TCRs were filtered to retain 1α1β or 2α1β for each clone.

### TCR repertoire analysis

TCRs were filtered to retain 1/2α or 1β; paired αβ cells consist of 1α1β or 2α1β. Clonotypes were defined as α (CDR3α amino acid + TRAV), β (CDRβ amino acid + TRBV) or paired αβ (CDR3α amino acid + TRAV + CDRβ amino acid + TRBV). Public clonotypes were defined as shared clonotypes between 2 or more patients. Circos plots were plotted using circlize v.0.4.12, showing paired TRAV-TRBV. All other plots were generated using ggplot2 and ggpubr v.0.4.0.

### RNAseq analysis

FASTQ files were generated from BCL files using bcl2fastq (v2.20.0) and were aligned to the hg19 genome using the bwa-MEM algorithm (v 0.7.15-r1140). Aligned reads were counted and count tables generated using featureCounts (v2.0.0) from the subread-2.0.0 package, using the hg19 annotation GTF file from ENSEMBL. Downstream analysis of the RNA counts was carried out in R (v4.2.0) using DESeq2[60] (v1.38.3). An exploratory analysis of the data were carried out by principal component analysis (PCA) after variance stabilising transformation.

### Proteomic analysis

Raw files were analysed in DIA-NN software (v8.0) as previously described[61]. Default settings were used as recommended. Briefly, for the library-free approach, a library was created from the human UniProt SwissProt database (downloaded 202102, containing 20,386 sequences) using deep learning. Trypsin was selected as the enzyme (1 missed cleavage), with carboamidomethylation of C as a fixed modification, oxidation of methionine as a variable modification and N-term M excision. Identification and quantification of raw data were performed against the in-silico library applying 1% FDR at the precursor level and match between runs (MBR). The DIA-NN 'report.proteingroup' matrix was further analysed using Perseus.

Protein intensities were log2-transformed and subsequently normalised by subtracting the median intensity by column. Biological replicates were grouped, and data were prefiltered, allowing three valid numbers in at least one biological group. Missing values were then imputed following a normal distribution. Imputed and normalised data were imported into R (v4.2.0). The mass spectrometry raw data included in this paper had been deposited to the Proteome eXchange Consortium via the PRIDE partner repository[62] with the dataset identifier PRD042469.

### Statistical analysis

$EC_{50}$ calculations were performed with GraphPad Prism 9, all other statistics were analyzed with IBM SPSS Statistics 27. Figures were made with ggplot2 in R (v4.2.0). Chi-squared test of independence was used to compare the ratio difference between two groups. Data distribution normality was examined with the Kolmogorov–Smirnov test. The Mann–Whitney $U$-test was employed to compare two groups, and the Kruskal–Wallis one-way ANOVA was used to compare three groups. Correlation analysis was performed using Spearman's rank correlation coefficient. $EC_{50}$ of T cell clones was calculated by using nonlinear regression with variable slope (four parameters) in a dose–response–stimulation model with GraphPad Prism. Statistical significance was set at $p < 0.05$, and all tests were two-tailed. For statistical analyses conducted using R, the MAST test was used to find DEGs between two conditions, adjusting for variation in batches and represented by volcano plots and violin plots. $P$ values were adjusted for multiple testing using the Benjamini–Hochberg method.

### TCR repertoire analysis of public datasets

Processed data were downloaded from GEO for GSE152522 (Meckiff et al.) and GSE162086 (Bacher et al.). As the Meckiff et al. data does not provide V gene usage, for Meckiff et al. cells were annotated as having a dominant TCRα if their CDR3 alpha sequence matched CIVR[A-Z] ANQAGTALIF, CIVRV[A-Z][Y/W]NFNKFYF, CVVN[A-Z][A-Z]SSNTGKLIF, CVNN[A-Z]GSSASKIIF or CA[A-Z][A-Z]NYGGSQGNLIF, allowing for any amino acid where "[A-Z]" is noted or allowing for either a 'Y' or 'W' at [Y/W]. For Bacher et al., as they provided both the CDR3 and V gene, we performed as above, but included the V gene in the matching, such that the alpha clonotype (CDR3 alpha + TRAV gene) had to match one of the following: CIVR[A-Z]ANQAGTALIF_TRAV26-1, CIVRV[A-Z][Y/W] NFNKFYF_TRAV26-1, CVVN[A-Z][A-Z]SSNTGKLIF_TRAV12-1, CVNN[A-Z] GSSASKIIF_TRAV12-1, or CA[A-Z][A-Z]NYGGSQGNLIF_TRAV35. All other cells were classed as having a non-dominant clonotype. The definition of disease severity was as reported by each manuscript.

For the prediction of HLA typing, we used arcasHLA. For Bacher et al., paired-end FASTQ files for the gene expression for each sample were downloaded from GSE162086 and used as input for arcasHLA. For Meckiff et al., due to eight samples having been multiplexed into each reaction, gene expression FASTQ files were downloaded from GSE152522, and we used vireoSNP to carry out genetic demultiplexing of the individual samples. Briefly, cellsnp-lite was used to call variants for each cell from the scRNA-seq data, using the provided list of 7.4 M human variants from the 1000 genome project with minor allele frequency (MAF) >0.05. Once the genotypes were called, vireo demultiplexed each sequencing run into individual FASTQ files, which were then used as input for arcasHLA.

The proportion of cells with a dominant and non-dominant TCR alpha clonotype between different disease severities was examined using the Chi-squared ($\chi^2$) test of independence. Results are reported as: $\chi^2$ (degrees of freedom, $N$ = sample size) = $\chi^2$ value, $p = p$ value.

### Reporting summary

Further information on research design is available in the Nature Portfolio Reporting Summary linked to this article.

## Data availability

The single-cell RNAseq data generated in this study have been deposited in the ArrayExpress database under accession code E-MTAB-14933. The mass spectrometry proteomics data have been deposited to the ProteomeXchange Consortium via the PRIDE partner repository with the dataset identifier PXD042469. Source data for all other figures are provided in a Source Data file. Source data are provided with this paper.

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

## Acknowledgements

We are grateful to all the participants for donating their samples and data for these analyses, and the research teams involved in the consenting, recruitment, and sampling of these participants. We thank K. Clark and S.-A. Clark from the WIMM Flow Cytometry facility for their help with cell sorting and the WIMM Sequencing facility for sequencing. This work is supported by the Chinese Academy of Medical Sciences (CAMS) Innovation Fund for Medical Sciences (CIFMS), China (grant number: 2024-I2M-2-001-1) (T.D., Y.P., X.Y., G.L., E.A., D.D., J.C.K., G.O., G.R.S., B.K. and R.F.); UK Medical Research Council (MRC) grant MR/Y015347/1 (T.D.), MR/R022011/1 (G.O.) and IMMPROVE - MR/Y004450/1 (T.D.); Z.Y., W.W., G.L. and C.L. were supported by China Scholarship Council. The study is also funded by the NIHR Oxford Biomedical Research Centre (G.O. and J.C.K.), Wellcome Trust Investigator Award (204969/Z/16/Z) (J.C.K.), Wellcome Trust Grants (090532/Z/09/Z and 203141/Z/16/Z) to core facilities Wellcome Centre for Human Genetics, Senior Investigator Award (G.O.) and Clinical Research Network (G.O.); Schmidt Futures (G.R.S.). The McKeating laboratory is funded by a Wellcome Investigator Award (IA) 200838/Z/16/Z. This work uses data provided by patients and collected by the NHS as part of their care and support #DataSavesLives. NIH Tetramer Facility provided $S_{166-180}$(CTFEYVSQPFLMDLE)-DPB1*04:01 monomer. Figure 1a was created with BioRender.com.

## Author contributions

T.D. conceptualised the project; T.D. and Y.P. designed and supervised T cell experiments; J.C.K. supervised bioinformatic analysis, A.M. supervised sample collection; G.L., Y.P., X.Y., Z.Y., D.D. and W.W. performed all T cell experiments and data analysis; E.A. performed single cell data analysis and bulk RNAseq analysis; P.A.C.W. and J.A.M. assisted with virus infection; T.R. performed HLA typing and next generation sequencing; I.V, B.K. and R.F performed proteomics experiments and initial analysis; J.W.F. provided MHC Class II Tetramers; J.C.K., A.J.M. and A.F. established clinical cohorts and collected clinical samples and data; C.W., K.C., P. Sopp, W.D., P. Supasa, C.L., J.M. and G.R.S. provided technical assistance and critical reagents; T.D., J.C.K. and Y.P. supervised data analysis, E.A., T.D., Y.P. and G.L. wrote the original draft; J.C.K., J.A.M. and G.O. reviewed and edited the manuscript and figures.

## Competing interests

The authors declare no competing interests.

## Additional information

[1]Chinese Academy of Medical Science (CAMS) Oxford Institute (COI), University of Oxford, Oxford, UK. [2]Centre for Translational Immunology, Nuffield Department of Medicine, University of Oxford, Oxford, UK. [3]Centre for Human Genetics, Nuffield Department of Medicine, University of Oxford, Oxford, UK. [4]Sequencing Facility, MRC Weatherall Institute of Molecular Medicine, University of Oxford, Oxford, UK. [5]Flow Cytometry Facility, MRC Weatherall Institute of Molecular Medicine, University of Oxford, Oxford, UK. [6]ProImmune Limited, Oxford, UK. [7]Target Discovery Institute, Centre for Medicines Discovery, Nuffield Department of Medicine, Oxford University, Oxford, UK. [8]Dengue Hemorrhagic Fever Research Unit, Office for Research and Development, Faculty of Medicine, Siriraj Hospital, Mahidol University, Bangkok, Thailand. [9]MRC Translational Immune Discovery Unit, MRC Weatherall Institute of Molecular Medicine, Radcliffe Department of Medicine, University of Oxford, Oxford, UK. [10]These authors contributed equally: Guihai Liu, Elie Antoun. [11]These authors jointly supervised this work: Alexander J. Mentzer, Julian C. Knight, Yanchun Peng, Tao Dong. ✉e-mail: tao.dong@ndm.ox.ac.uk

