## [Transparent Peer Review file · Nature Communications]

Long-persisting SARS-CoV-2 spike-specific CD4+ T cells associated with mild disease and increased cytotoxicity post COVID-19

Corresponding Author: Professor Tao Dong

Version 0:

Reviewer comments:

Reviewer #1

(Remarks to the Author)

The authors have modified the manuscript and, as such, improved it. I do have a couple of questions remaining:

Mild and severe outcomes of COVID-19 need to be defined.

It is key to know the infection and vaccination history of the patients over the 4 years, and analyze spike-specific CD4+ T cell responses according to the number of infections/vaccinations. It is possible that the public TCRs persisted in the patients who had repeated COVID-19 and/or COVID vaccination.

The authors found prevalence of TRBV24-1 at 1-3 months and dominance of TRBV6-1 at 3-4 years for S751-specific CD4+ T cells. Similarly, TRBV3-1 was found at 3-4 years for S866-specific CD4+ T cells. Was this finding similar across all the participants, or are those results impacted by a particular individual with skewed TCR usage?

Would conservation of S166-180, S751-765 and S866-880 across SARS-CoV-2 variants and seasonal coronaviruses potentially affect repeated recall of these spike-specific CD4+ T cells, thus their TCR repertoire, survival and cytokine profile?

Reviewer #2

(Remarks to the Author)

This revision by Liu, Antoun, and colleagues addresses many of the issues raised in the initial round of peer review. The revised manuscript is substantially clearer than the initial version. However, the experiment outlined in Figure 3 that evaluates putative associations between five TRA motifs and disease severity still requires further explanation.

1) When checking the GEO database for both manuscripts reportedly included in the analysis for Figure 3a-3c, this reviewer is only able to locate 47 VDJ files. The text on line 201 of the present manuscript indicates that 51 COVID-19 patients are included in the analysis. Please show a table that includes the subject study number and the severity of disease used in this analysis for each included study participant from the two studies. The N of subjects from each severity group from each manuscript used in the presented analysis should be included. The authors of Meckiff et al. use the terminology ICU rather than ITU to indicate severity - please use the original terminology from the parent manuscript in the figures and tables discussing this experiment.

2) There are substantial limitations to the unknown HLA-DRB1*15:01 and -DPB1*04:01 haplotype frequency of the participants of varying disease severity in this analysis. Unbalance in the distribution of HLA-DRB1*15:01 in the outcome groups, while interesting in itself, could lead to bias in the presented results. Please break down the analysis of dominant and non-dominant TRA clonotypes in the total combined cohort (panel 3c) for each of the three analyzed epitopes. Do one

(S166-180) or two (S751-765 and S866-880) epitopes drive the difference in disease severity proportion?

3) The strongly-worded discussion sentence included in the results section lines 210-212 is out of place in the results section and is also overstated given the limitations of this experiment.

4) The limitations of the findings in Figure 3 are not described in the current version of the discussion section. These limitations should be discussed.

5) The methods section still does not include information about the analyses performed in Figure 3. These methods must be included in the methods section.

6) Figure 4c and 4d - it would improve clarity to coordinate the colors of these two panels of the figure - blue indicates "both timepoints" in panel c, yet indicates "1-3 month only" in panel d.

Version 1:

Reviewer comments:

Reviewer #1

(Remarks to the Author)

I thank the Authors for addressing the comments. I have no further concerns.

Reviewer #2

(Remarks to the Author)

The authors have addressed all concerns in the revised manuscript.

Point to point responses to Reviewers' comments:

Reviewer #1 (Remarks to the Author):

The authors have modified the manuscript and, as such, improved it. I do have a couple of questions remaining:

C1: Mild and severe outcomes of COVID-19 need to be defined.

R1: We appreciate this comment. We have now added the definition of mild and severe outcomes of COVID-19 in the methods on lines 501-502.

"In brief, the degree of severity was identified as mild, severe or critical infection, according to recommendations from the World Health Organization."

C2: It is key to know the infection and vaccination history of the patients over the 4 years, and analyze spike-specific CD4+ T cell responses according to the number of infections/vaccinations. It is possible that the public TCRs persisted in the patients who had repeated COVID-19 and/or COVID vaccination.

R2: We fully agree with this comment. Vaccination history of patients with 3-4-year follow-up is shown in Supplementary Table 1, with all participants having had at least 3 vaccine doses before the sampling at 3-4 years post initial infection. Unfortunately, detailed infection records between the initial infection and follow-up time point are unavailable. However, given the widespread prevalence of Omicron during 2022–2023, it is reasonable to assume that all donors experienced at least two infections—an initial infection and a subsequent one with an Omicron variant.

Regarding the potential impact of repeated infections or vaccinations on the persistence of public TCRs, we analysed the relationship between the number of vaccine doses and the proportion of public TCRα clonotypes at 3–4 years post-infection. We did not observe a significant correlation ($p = 0.9049$; see figure below). Moreover, we found no evidence of preferential retention of public TCRs in specific individuals; all participants exhibited cells with public TCRα clonotypes. This information is now added to the revised manuscript on lines 183-187.

C3: The authors found prevalence of TRBV24-1 at 1-3 months and dominance of TRBV6-1 at 3-4 years for S751-specific CD4+ T cells. Similarly, TRBV3-1 was found at 3-4 years for S866-specific CD4+ T cells. Was this finding similar across all the participants, or are those results impacted by a particular individual with skewed TCR usage?

R3: We appreciate this comment. Yes, this finding was similar across all the participants, and we did not see any impact of one particular individual. We observed four individuals at 1-3 months having dominant TRBV24-1 and four individuals at 3-4 years having dominant TRBV6-1 specific to epitope S751; and three individuals at 3-4 years having dominant TRBV3-1 specific to epitope S866. These findings suggest that the observed pairing patterns are not

driven by the skewed TCR usage in a particular individual. We have added this important point to the revised manuscript on lines 160-161.

C4: Would conservation of S166-180, S751-765 and S866-880 across SARS-CoV-2 variants and seasonal coronaviruses potentially affect repeated recall of these spike-specific CD4+ T cells, thus their TCR repertoire, survival and cytokine profile?

R4: This is a very good point. We believe the seasonal coronaviruses are unlikely to have affected repeated recall of the spike-specific CD4+ T cells, however variants may have. The sequences of S166-180, S751-765 and S866-880 epitopes show very little similarity to those of seasonal coronaviruses as shown below, suggesting that the impact of seasonal coronaviruses on these spike-specific CD4+ T cells is likely minimal. The three epitopes are highly conserved across SARS-CoV-2 variants, so spike-specific CD4+ T cells targeting these epitopes are likely have been repeatedly stimulated by reinfections or multiple rounds of vaccination. These repeated stimulations may have affected their TCR repertoire, phenotype and function as observed in our data from the samples at 3-4 years after initial primary infection. We have added this point to the revised manuscript in the discussion section, on lines 431-437.

		S166-180		S751-765		S866-880
SARS-Cov-2 (MN908947)	-166	CTFEYVSQPFLMDLE	-751	NLLLQYGSFCTQLNR	-866	TDEMIAQYTSALLAG
HCov 229E (KU291448)	-113	-----DVIRYNIN	-619	E..K..T.A.KTIED	-739	DA.RM.M..GS.IG.
HCov OC43 (MN306036)	-202	.LYK---RN.TY.VN	-834	LQ.VE.....DNI.A	-959	SENQ.SG..L.ATSA
HCov NL63 (NC_005831)	-292	-----VDVMRYN.N	-802	...K..T.A.KTIED	-922	DA.RM.M..GS.IG.
HCov HKU1 (NC_006577)	-183	.L.K---KN.TYNVS	-831	D..SE..T..DNI.S	-955	SESQ.SG..T.ATVA

"-" represents depletion, "." represents consistent amino acid

Reviewer #2 (Remarks to the Author):

This revision by Liu, Antoun, and colleagues addresses many of the issues raised in the initial round of peer review. The revised manuscript is substantially clearer than the initial version. However, the experiment outlined in Figure 3 that evaluates putative associations between five TRA motifs and disease severity still requires further explanation.

C1: When checking the GEO database for both manuscripts reportedly included in the analysis for Figure 3a-3c, this reviewer is only able to locate 47 VDJ files. The text on line 201 of the present manuscript indicates that 51 COVID-19 patients are included in the analysis. Please show a table that includes the subject study number and the severity of disease used in this analysis for each included study participant from the two studies. The N of subjects from each severity group from each manuscript used in the presented analysis should be included. The authors of Meckiff et al. use the terminology ICU rather than ITU to indicate severity - please use the original terminology from the parent manuscript in the figures and tables discussing this experiment.

R1: We appreciate the comments. GSE162086 (Bacher et al.) have 20 VDJ files in their repository, of which we used 14 files corresponding to the patients who had COVID19 (not healthy patients) and for which severity was available. GSE152522 (Meckiff et al.) have 27 TCR files in their repository, however, their study design involved hashtagging up to 8 samples from different patients into the same reaction (as mentioned in their publication), and as such each of those files has more than 1 sample/patient. As Meckiff et al. do not provide information on which hashing antibodies were used for which samples, we used vireo to genetically demultiplex the data for HLA prediction. However, for the purposes of this analysis, we used the already demultiplexed and processed data available in the annotation files

(GSE152522_cd4t0n6_annotation.txt.gz and GSE152522_cd4t6_annotation.txt.gz files). From their publication, they mention they sequenced >100000 cells from 53 individuals, despite only have 27 TCR files in their GEO repository. We used the data from 38 of those individuals. So, 52 COVID-19 patients are included in the analysis.

We have now included below a table of the number of subjects from each manuscript in each severity group and added it as Supplementary Table 4. HLA types as predicted by arcasHLA from the gene expression FASTQ files are also included.

For Meckiff et al. due to low sequencing depth as a result of low numbers of cells for certain patients, arcasHLA was unable to predict the HLA type of the individuals, due to the low number of reads mapping to the HLA locus. These are shown as empty in the table below.

Meckiff et al.						Bacher et al.					
ID	Status	DPB1-1	DPB1-2	DRB1-1	DRB1-2	ID	Status	DPB1-1	DPB1-2	DRB1-1	DRB1-2
P01	Ward			14:04:01	07:01:01	J10535	mild	05:01:01	04:01:01	01:01:01	03:01:01
P03	Ward			14:54:01	15:01:01	J10886	mild	02:01:02	04:01:01	07:01:01	08:02:01
P04	Ward					J10888	mild	04:01:01	04:02:01	07:01:01	11:01:01
P05	ICU					J14205	mild	02:01:02	04:01:01	04:03:01	01:01:01
P06	Ward			01:01:01	03:01:01	J15893	mild	04:01:01	04:01:01	08:01:01	11:01:01
P07	Ward			14:54:01	07:01:01	J09835	non-hospitalised	04:01:01	04:02:01	07:01:01	15:01:01
P0901	Ward	04:01:01	04:01:01	15:01:01	16:01:01	J09836	non-hospitalised	02:01:02	04:01:01	11:01:01	15:01:01
P10	ICU	18:01:01	01:01:01	07:01:01	15:03:01	J10624	non-hospitalised	04:01:01	04:01:01	15:01:01	15:01:01
P12	Ward	1321:01:00	04:01:01	15:01:01	11:04:01	J10625	non-hospitalised	02:01:02	04:01:01	01:01:01	15:01:01
P15	Ward					J11689	non-hospitalised	04:01:01	01:01:01	03:01:01	15:01:01
P16	Ward	09:01:01	04:02:01	07:01:01	15:02:01	J15890	non-hospitalised	02:01:02	04:01:01	04:01:01	13:01:01
P17	Ward	1321:01:00	03:01:01	01:01:01	13:02:01	J10887	severe	14:01:01	04:02:01	14:54:01	04:01:01
P18	Ward			04:01:01	07:01:01	J14204	severe	02:01:02	02:01:02	13:02:01	11:04:01
P19	Ward			15:01:01	14:54:01	J21854	severe	04:01:01	04:02:01	04:01:01	13:02:01
P20	ICU	04:01:01	04:02:01	11:03:01	13:01:01						
P22	mild	1321:01:00	04:01:01	04:03:01	11:04:01						
P24	ICU	04:02:01	1068:01:00	04:01:01	08:01:01						
P25	mild			07:01:01	04:04:01						
P26	mild										
P27	ICU										
P29	mild										
P30	mild			15:01:01	14:54:01						
P31	mild										
P32	mild										
P37	mild										
P40	mild			01:01:01	04:01:01						
P42	ICU			15:06:01	01:01:01						
P43	ICU			13:36	04:05:01						
P44	mild	04:01:01	10:01:02	12:01:01	11:01:01						
P45	mild										
P46	ICU	01:01:01	06:01:01	07:01:01	03:01:01						
P47	mild										
P49	ICU			15:06:01	03:01:01						
P57	mild										
P61	mild										
P64	mild										
P66	mild			09:01:02	03:01:01						

Regarding the terminology ICU or ITU, the Meckiff et al. manuscript does use ICU to indicate severity, however in the annotation files listed above from which the TCR data was obtained, the severity is defined as ITU. We have now changed ITU to ICU in Figure 3a below and the supplementary table above.

C2: There are substantial limitations to the unknown HLA-DRB1*15:01 and -DPB1*04:01 haplotype frequency of the participants of varying disease severity in this analysis. Unbalance in the distribution of HLA-DRB1*15:01 in the outcome groups, while interesting in itself, could lead to bias in the presented results. Please break down the analysis of dominant and non-dominant TRA clonotypes in the total combined cohort (panel 3c) for each of the three analyzed epitopes. Do one (S166-180) or two (S751-765 and S866-880) epitopes drive the difference in disease severity proportion?

R2: Thanks for this comment. We have now broken down the analysis of dominant and non-dominant TRA clonotypes in the combined cohort for one (S166-180) or two (S751-765 and S866-880) epitopes. The dominant TRA clonotypes of the S166-180 epitope in HLA-DPB1*04:01+ individuals are associated with mild COVID-19 disease (left figure), and the dominant TRA clonotypes of S751-765 and S866-880 epitopes in HLA-DRB1*15:01+ participants are correlated with non-hospitalised COVID-19 (right figure). Therefore, the difference in disease severity proportion was not driven by one (S166-180) or two (S751-765 and S866-880) epitopes. We have now included these figures in Extended Data Fig. 3 and added this to the text in the results section, on lines 213-223.

C3: The strongly-worded discussion sentence included in the results section lines 210-212 is out of place in the results section and is also overstated given the limitations of this experiment.

R3: Thanks for this comment. We have revised the discussion sentence to “In summary, these data suggest that these dominant TCR α clonotypes play an important role in protecting individuals from developing severe disease.”, on lines 223-225.

C4: The limitations of the findings in Figure 3 are not described in the current version of the discussion section. These limitations should be discussed.

R4: We appreciate this comment and have added the limitations to the discussion section on lines 403-405.

“Given the limited number of participants in the two studies and the absence of HLA-typing for some individuals in Meckiff et al. study²⁹, these findings need to be further validated in larger cohorts.”

C5: The methods section still does not include information about the analyses performed in Figure 3. These methods must be included in the methods section.

R5: We apologise that we did not include these analysis methods in the methods section. We have now added them to the methods on lines 747-773.

“Processed data were downloaded from GEO for GSE152522 (Meckiff et al.) and GSE162086 (Bacher et al.). As the Meckiff et al. data does not provide V gene usage, for Meckiff et al. cells were annotated as having a dominant TCR α if their CDR3 alpha sequence matched CIVR[A-Z]ANQAGTALIF, CIVRV[A-Z][Y/W]NFNKFYF, CVVN[A-Z][A-Z]SSNTGKLIF, CVNN[A-Z]GSSASKIIF or CA[A-Z][A-Z]NYGGSQGNLIF, allowing for any amino acid where “[A-Z]” is noted or allowing for either a “Y” or “W” at [Y/W]. For Bacher et al., as they provided both the CDR3 and V gene, we performed as above, but included the V gene in the matching, such that the alpha clonotype (CDR3 alpha + TRAV gene) had to match one of the following: CIVR[A-Z]ANQAGTALIF_TRAV26-1, CIVRV[A-Z][Y/W]NFNKFYF_TRAV26-1, CVVN[A-Z][A-Z]SSNTGKLIF_TRAV12-1, CVNN[A-Z]GSSASKIIF_TRAV12-1, or CA[A-Z][A-Z]NYGGSQGNLIF_TRAV35. All other cells were classed as having a non-dominant clonotype. Definition of disease severity was as reported by each manuscript.

For prediction of HLA typing, we used arcasHLA. For Bacher et al. paired end FASTQ files for the gene expression for each sample were downloaded from GSE162086 and used as input for arcasHLA. For Meckiff et al., due to 8 samples having been multiplexed into each reaction, gene expression FASTQ files were downloaded from GSE152522 and we used vireoSNP to carry out genetic demultiplexing of the individual samples. Briefly, cellsnp-lite was used to call variants for each cell from the scRNAseq data, using the provided list of 7.4M Human variants from 1000 genome project with minor allele frequency (MAF) > 0.05. Once the genotypes were called, vireo demultiplexed each sequencing run into individual FASTQ files, which were then used as input for arcasHLA.

Proportion of cells with a dominant and non-dominant TCR alpha clonotype between different disease severities was examined using the Chi-squared (χ^2) test of independence. Results are reported as: χ^2 (degrees of freedom, N = sample size) = chi-square value, p = p-value.”

C6: Figure 4c and 4d - it would improve clarity to coordinate the colors of these two panels of the figure - blue indicates “both timepoints” in panel c, yet indicates “1-3 month only” in panel d.

R6: We appreciate this comment. We have coordinated the colours of Figure 4c and d as shown below.

C

d